# Emission Monitoring Mobile Experiment (EMME): an overview and first results of the St. Petersburg megacity campaign-2019

Maria V. Makarova[1], Carlos Alberti[2], Dmitry V. Ionov[1], Frank Hase[2], Stefani C. Foka[1], Thomas Blumenstock[2], Thorsten Warneke[3], Yana A. Virolainen[1], Vladimir S. Kostsov[1], Matthias Frey[4], Anatoly V. Poberovskii[1], Yuri M. Timofeyev[1], Nina N. Paramonova[6], Kristina A. Volkova[1], Nikita A. Zaitsev[1], Egor Y. Biryukov[1], Sergey I. Osipov[1], Boris K. Makarov[5], Alexander V. Polyakov[1], Viktor M. Ivakhov[6], Hamud Kh. Imhasin[1], Eugene F. Mikhailov[1]

[1] Department of Atmospheric Physics, Faculty of Physics, St. Petersburg State University, Russia

[2] Institute of Meteorology and Climate Research IMK-ASF, Karlsruhe Institute of Technology, Karlsruhe, Germany

[3] University of Bremen, Germany

[4] National Institute for Environmental Studies, Japan

[5] Institute of Nuclear Power Engineering, Peter the Great St. Petersburg Polytechnic University, Russia

[6] Voeikov Main Geophysical Observatory, St. Petersburg, Russia

*Correspondence to: Maria V. Makarova (m.makarova@spbu.ru), Frank Hase (Frank.Hase@kit.edu), and Dmitry V. Ionov (d.ionov@spbu.ru)*

**Abstract.** Global climate change is one of the most important scientific, societal and economic contemporary challenges. Fundamental understanding of the major processes driving climate change is the key problem which is to be solved not only on a global but also on regional scales. The accuracy of regional climate modelling depends on a number of factors. One of these factors is the adequate and comprehensive information on the anthropogenic impact which is highest in industrial regions and areas with dense population – modern megacities. Megacities are not only "heat islands", but also significant sources of emissions of various substances into the atmosphere, including greenhouse and reactive gases. In 2019, the mobile experiment EMME (Emission Monitoring Mobile Experiment) was conducted within the St. Petersburg agglomeration (Russia) aiming to estimate the emission intensity of greenhouse ($CO_2$, $CH_4$) and reactive (CO, $NO_x$) gases for St. Petersburg which is the largest Northern megacity. St. Petersburg State University (Russia), Karlsruhe Institute of Technology (Germany) and the University of Bremen (Germany) jointly ran this experiment. The core instruments of the campaign were two portable FTIR spectrometers Bruker EM27/SUN which were used for ground-based remote sensing measurements of the total column amount of $CO_2$, $CH_4$ and CO at upwind and downwind locations on the opposite sides of the city. The $NO_2$ tropospheric column amount was observed along a circular highway around the city by continuous mobile measurements of scattered solar visible radiation with OceanOptics HR4000 spectrometer using the DOAS technique. Simultaneously, air

samples were collected in air bags for subsequent laboratory analysis. The air samples were taken at the locations of FTIR observations at the ground level and also at altitudes of about hundred meters when airbags were lifted by a kite (in case of suitable landscape and favourable wind conditions). The entire campaign consisted of 11 mostly cloudless days of measurements in March-April 2019. Planning of measurements for each day included the determination of optimal location for FTIR spectrometers based on weather forecasts combined with the numerical modelling of the pollution transport in the megacity area. The real-time corrections of the FTIR operation sites were performed depending on the actual evolution of the megacity $NO_x$ plume as detected by the mobile DOAS observations. The estimates of the St. Petersburg emission intensities for the considered greenhouse and reactive gases were obtained by coupling a box model and the results of the EMME observational campaign using the mass balance approach. The $CO_2$ emission flux for St. Petersburg as an area source was estimated as $89\pm28$ kt km$^{-2}$ yr$^{-1}$ which is two times higher than the corresponding value in the EDGAR database. The experiment revealed the $CH_4$ emission flux of $135\pm68$ t km$^{-2}$ yr$^{-1}$ which is about one order of magnitude greater than the value reported by the official inventories of St. Petersburg emissions (~25 t km$^{-2}$ yr$^{-1}$ for 2017). At the same time, for the urban territory of St. Petersburg, both the EMME experiment and the official inventories for 2017 give similar results for the CO anthropogenic flux ($251\pm104$ t km$^{-2}$ yr$^{-1}$ vs. 410 t km$^{-2}$ yr$^{-1}$) and for the $NO_x$ anthropogenic flux ($66\pm28$ t km$^{-2}$ yr$^{-1}$ vs. 69 t km$^{-2}$ yr$^{-1}$).

**Keywords:** ground-based remote sensing, portable spectrometers, FTIR spectroscopy, DOAS technique, mobile experiments, trace gas retrieval, greenhouse gases, reactive gases, anthropogenic emissions in megacities, transport modelling of air pollutants

## 1 Introduction

Global climate change is one of the most important scientific, societal and economic contemporary challenges. Fundamental understanding of the major processes driving climate change is the key problem which is to be solved not only on a global but also on regional scales (IPCC, 2013; WMO Greenhouse Gas Bulletin, 2018). The accuracy of regional climate modelling depends on a number of factors. One of these factors is the adequate and comprehensive information on the anthropogenic impact which is highest in industrial regions and areas with dense population - modern agglomerations and megacities. Agglomerations and megacities are not only "heat islands", but also significant sources of emissions of various substances into the atmosphere, including greenhouse and reactive gases (Zinchenko et al., 2002; Wunch et al., 2009; Ammoura et al., 2014; Hase et al., 2015; Turner et al., 2015; Viatte et al., 2017). Estimating emission intensity for industrial areas and cities requires precise measurements of gas composition in the troposphere with a high horizontal resolution on a regional scale. Existing ground-based observational networks, in particular ESRL (ESRL, 2019), ICOS (ICOS, 2020), NDACC (NDACC,

2019) and TCCON (TCCON, 2019), are mainly focused on detecting the background concentrations of the greenhouse gases. Most of observational stations are sparsely distributed and located relatively far from industrial and highly populated areas. Portable Fourier Transform InfraRed (FTIR) spectrometers EM27/SUN (Gisi et al., 2012, Frey et al., 2015) are very promising instruments for the detection and quantification of the emissions of greenhouse gases from mesoscale area sources like cities or industrial areas (Hase et al., 2015; Chen et al., 2016). The data provided by these instruments are less affected by the vertical exchange processes than the data obtained from in situ measurements. Also, in contrast to current space-based sensors, the ground-based portable FTIR spectrometer data are essentially unaffected by the aerosol burden transported by the pollution plume.

The quantification of the gas fluxes from the sources located on the earth's surface can be carried out using various methods: the "forward" and "inverse" modelling (Maksyutov et al., 2013; Turner et al., 2015), the eddy covariance method (Helfter et al., 2011; Hiller et al., 2014a), the mass balance approach (Zimnoch et al., 2010; Strong et al., 2011, Hiller et al., 2014a), and the technique based on the radon measurements (Lopez et al., 2015). Depending on a method, the spatial coverage of investigated sources can vary from the local (for example, in the case of eddy covariance) to the meso- and the global scales (the assimilation of satellite data in atmospheric models). Each of these approaches has its own set of unique advantages and limitations depending on specific spatial and/or temporal scales. Therefore the efficacy and accuracy of many of these methods remain the subject of scientific debates (Cambaliza et al., 2014; Hiller et al., 2014a). Often, combinations of these methods can yield reduced uncertainty of target parameters, at the same time combining of different techniques often requires special field campaigns and comprehensive analysis (Hiller et al., 2014a; Hiller et al., 2014b).

Recently, several studies were performed with the goal to estimate the emissions of industrial regions and cities by means of ground-based mobile measurements of tropospheric gaseous composition using the FTIR and DOAS technique. Hase et al. (2015) and Zhao et al. (2019) applied portable FTIR spectrometers for detecting greenhouse gas emissions of the major city Berlin. In these studies, five portable EM27/SUN spectrometers were used for the accurate and precise observations of column-averaged abundances of $CO_2$ and $CH_4$ around the major city Berlin. It has been demonstrated that the $CO_2$ emissions of Berlin can be clearly identified in the observations. Chen et al. (2016) developed and used differential column methodology (downwind-minus-upwind column differences) for the evaluation of $CH_4$ emissions from dairy farms in the Chino area. Vogel et al. (2019) investigated the Paris megacity emissions of $CO_2$ by coupling the COCCON observations and atmospheric transport model framework (CHIMERE-CAMS) simulations. Luther et al. (2019) explored the feasibility of estimating $CH_4$ emissions for individual coal mine ventilation shafts and groups of shafts. They measured column-averaged dry-air mole fractions of methane $XCH_4$ by the FTIR spectrometer Bruker EM27/SUN which was installed on a truck moving through the $CH_4$ plumes in the Upper Silesian Coal Basin while driving in stop-and-go patterns. De Foy et al. (2007), Mellqvist et al. (2010), Johansson et al. (2014), and Kille et al. (2017) have applied mobile FTIR (Solar

Occultation Flux technique) and mobile DOAS techniques to the large scale flux measurements. Babenhauserheide et al. (2020) estimated $CO_2$ emissions from Tokyo using the long-term statistical analysis of $XCO_2$ amounts measured at the Tsukuba TCCON site located near Tokyo.

The motivation of the present study originated from the fact that the number of observational stations for greenhouse gas monitoring on the territory of Russia is very limited and there are considerable uncertainties of the greenhouse gas flux estimations for the natural and anthropogenic sources in Russia. St. Petersburg is the second largest megacity in Russia with the population of 5 million and, besides, it is the northernmost city in the world with the population of over one million people. The goal of the present study was to estimate the emissions of greenhouse ($CO_2$, $CH_4$) and reactive (CO, $NO_x$) gases from St. Petersburg by means of mobile remote-sensing techniques and direct in situ measurements. The study was based on the observational campaign EMME-2019 (Emission Monitoring Mobile Experiment) which was performed in March-April 2019 on the territory of the St. Petersburg agglomeration. St. Petersburg State University (Russia), Karlsruhe Institute of Technology (Germany) and the University of Bremen (Germany) jointly ran this experiment in the frame of the International project VERIFY (VERIFY, 2019). The idea and the methodology of EMME experiment were based mainly on the studies by Hase et al. (2015), Ionov and Poberovskii (2015), Chen et al. (2016) and Viatte et al. (2017).

## 2 Concept of EMME, instruments and the experiment planning

The concept of EMME is based on remote measurements of the total column amount of $CO_2$, $CH_4$ and CO from two mobile platforms located inside and outside the city plume (usually at upwind and downwind locations on the opposite sides of the city of St. Petersburg) combined with the mobile circular measurements of tropospheric column amount of $NO_2$ from the third mobile platform moving in a non-stop mode, the latter measurements are used for the real-time control of the megacity plume evolution. The simplified illustration of the concept is given in Fig. 1. The experiment requires clear-sky conditions since the instruments for remote sensing measure direct and scattered solar radiation. The ancillary measurements include control of the meteorological parameters and sampling of air portions at the locations inside and outside the city plume for subsequent laboratory analysis of concentrations of target gases. In order to assess the intensity of gas emissions by St. Petersburg, the mass-balance approach is applied to the measurement data. The principal feature of EMME is its integrated character: several different instruments are used, and additionally, the planning of the field experiment and data processing are performed with the help of numerical modelling of the transport of the megacity pollution plume.

The core instruments of the campaign are two portable FTIR spectrometers Bruker EM27/SUN (Gisi et al., 2012; Frey et al., 2015, Hase et al., 2016) which are used for ground-based remote sensing measurements of total column amount of $CO_2$, $CH_4$ and CO. The EM27/SUN instrument has a sun-tracking system and registers direct infrared solar radiation. The

FTIR spectrometers are transported by cars to the measurement locations where they are unloaded and installed outside. The geographic coordinates are registered by the GNSS (Global Navigation Satellite System) sensor. A detached car battery with an inverter is used as a power supply which ensures about 3 h operation time. Under cold weather conditions, the instruments are covered by electric heating blankets. The integration time for a single spectrum constitutes about 1 min. Within this period, about 10 interferograms are registered and averaged, and then the corresponding spectrum is recorded.

The tropospheric $NO_2$ column is derived from measurements of the scattered solar radiation in the zenith direction by the portable automatic spectrometer OceanOptics HR4000. This spectrometer is mounted on board of a car and connected to a portable computer to ensure uninterruptible recording of spectra. Measurements are fully automatic while the car is moving. The location of the car is controlled by the GNSS sensor and is routinely recorded by the onboard computer for instant referencing of the results of measurements to the car route. The sampling period of time (time of exposure) for single spectrum is calculated by the software tool accounting for illumination conditions and constitutes about 60 ms on average for the observations at about noon. Recording of spectra is done every 1 min, all single spectra obtained within this period are coadded. Thus, each final measurement is the mean of about 1000 instant spectra. The route includes the entire city ringway (the highway around St. Petersburg), therefore the main emission sources are inside the route and the position of the megacity plume can be detected with high accuracy. The described approach and the DOAS mobile experiment specific design have been implemented previously at St. Petersburg and the results have been published by Ionov and Poberovskii (2012, 2015, 2017, 2019).

Air samples were collected at the locations of both FTIR spectrometers in two air bags: when FTIR measurements started (the first bag) and before completion of FTIR measurements (the second bag). Each bag was a 25-liter Tedlar bag, sampled for about 40 min. In case of suitable weather and landscape conditions at the location of one of the FTIR spectrometers, sampling bags were lifted by a kite to an altitude of about 100 m. The laboratory analysis of the air samples was performed with the help of gas analysers. Gas analyser Los Gatos Research GGA 24r-EP was used for measuring volume mixing ratio (vmr) of $CH_4$, $CO_2$ and $H_2O$. Gas analyser Los Gatos Research CO 23r was used for measuring vmr of CO and $H_2O$. The concentration of NO and $NO_2$ ($NO_x$) was measured by gas analyser ThermoScientific 42i-TL.

For the monitoring of meteorological parameters, two weather stations and the microwave radiometer RPG-HATPRO were used. One portable weather station was operating either at upwind or at the downwind location of FTIR spectrometers. The atmospheric pressure measurements were performed at both up- and downwind locations. The second stationary weather station was operating on the roof of the building (56 m a.s.l.) of the Institute of Physics of St. Petersburg State University (SPbU) located about 25 km west from the city centre. The RPG-HATPRO radiometer was operating also on the roof of this building and delivered information on the temperature and humidity vertical profiles together with the information on the cloud liquid water path (Kostsov, 2015; Kostsov et al., 2018).

The essential part of EMME was the preparatory stage which lasted for three months before the start of the campaign. During this stage the optimal set of FTIR measurement locations in the close vicinity of the St. Petersburg ringway was determined accounting for several criteria. First, this set of locations should have had sufficient spatial density to ensure the possibility to perform up- and down-wind FTIR measurements for practically any wind directions. Second, every location should have been convenient for car parking in the ringway proximity, and for installation of the instruments. We tried to choose the locations at a certain distance from the highway and roads with intensive traffic in order to avoid contamination of air by local sources. The set of FTIR measurement locations around the St. Petersburg agglomeration which was chosen during the preparatory stage is shown in Fig.2. It should be emphasized that during the preparatory stage a kind of rehearsal was carried out. This rehearsal has helped to reveal how time consuming the following processes are: loading the equipment on cars at the Institute of Physics, unloading the equipment at a measurement location, setting up and tuning the instruments for data acquisition. This information is critical for understanding whether it is possible to reach the desired up- and down-wind locations in proper time by different crews and to start simultaneous FTIR measurements.

Special attention was paid to planning of the experiment a day before. We analysed the weather forecasts presented by different sources with special attention to cloud cover and wind direction. Mainly, we used the cloud maps from https://www.msn.com (last access 12 November 2019). In order to determine FTIR measurement locations for specific day, we made a forecast of the megacity plume using the HYSPLIT (HYbrid Single-Particle Lagrangian Integrated Trajectories) model (Draxler and Hess, 1998; Stein et al., 2015). In addition, in the morning of a measurement day we monitored the cloud cover using web cameras which operated nearby the planned measurement locations.

**3 Overview of the 2019 campaign**

The EMME field campaign in 2019 consisted of 11 days of measurements in March-April. Table A1 (see Appendix A) presents daily information on the location of FTIR spectrometers during the campaign, FTIR spectrometer identifier, number of bags of air samples, flight of a kite and air sampling altitude. Below, we refer to the two Fourier Transform Spectrometers (FTS) as FTS#80 and FTS#84. In Table A2 (please, see Appendix A) we collect the main characteristics of weather conditions for each measurement day. The satellite images of cloud cover detected by the MODIS satellite instrument in the vicinity of St. Petersburg are presented in Fig. A1 (see Appendix A). They confirm daytime clear sky conditions for the duration of the campaign, except the day of April 30, when the altocumulus translucidus clouds started to develop.

During the EMME-2019 we implemented two types of field experiment setup regarding the position of FTIR spectrometers relative to the dominant air flow (wind) direction:

- for most of the days of observations (ten of the eleven), FTIR spectrometers were installed along the wind direction line -
180    in up- and downwind locations on the opposite sides of the city of St. Petersburg (Fig.1, locations #1 and #2);

- for 16 April – the cross sectional setup was implemented. FTIR spectrometers were located on the line which is nearly perpendicular to the dominant wind direction line (not shown in Fig.1).

In order to forecast the spatial distribution of urban air pollution on each day of campaign observations, we used the HYSPLIT model. Following our previous experience of simulating the dispersion of urban contamination from 185    St. Petersburg, the $NO_2$ content in the lower troposphere was set as a tracer of the polluted air mass distribution (Ionov and Poberovskii, 2019). This numerical modelling was done by means of the dispersion module within the offline version of HYSPLIT. It allowed performing the 3D simulation of the generation and dispersion of $NO_2$ plume from a set of given sources of anthropogenic $NO_x$ emission. The model was configured in the same way as in our early studies (Ionov and Poberovskii, 2012; Ionov and Poberovskii, 2015; Ionov and Poberovskii, 2017). Similar to the most recent study by Ionov 190    and Poberovskii (2019), the $NO_x$ emissions were specified according to the official municipal inventory of emission sources. The HYSPLIT grid domain was set with the centre at 58.20ºN and 30.75ºE, the grid spacing (horizontal spatial resolution) of 0.05º latitude and longitude, and the grid span of 6.8º latitude and 14.1º longitude. The vertical grid consisted of 10 levels with the tops at 1, 25, 50, 100, 150, 250, 350, 500, 1000 and 1500 m. The forecast meteorology data (vertical distributions of the horizontal and vertical wind components, temperature, pressure, etc.) were taken from the National Centers for 195    Environmental Prediction Global Forecast System (NCEP GFS, ftp://arlftp.arlhq.noaa.gov/forecast) on the 1º×1º latitude×longitude spatial grid. The maps of the $NO_2$ plume, simulated by the HYSPLIT model for 13:00 local time on each day of campaign observations, are presented in Fig. 3. Colour scale represents the spatial distribution of $NO_2$ column amount integrated within the boundary layer (~1500 m). An animated version of such a forecast, showing the plume evolution, was generated and shared among the campaign staff ~12 hours before each day of planned observations (an example of the 200    animated forecast for 6 April 2019 is available at https://youtu.be/rgtq6JLPhig, last access 2 March 2020).

Based on the plume evolution forecasts, the optimal pair of the FTIR spectrometer locations for the upcoming day of measurements was chosen. This approach to planning of the city campaign was implemented during 11 days of EMME-2019, and the necessity to change the location of the FTIR spectrometers occurred only once, on April 18. For this day, the real-time information on the $NO_2$ tropospheric column (TrC) acquired along the ringroad by the crew #3 using 205    mobile DOAS observations showed that the actual location of the most polluted city plume area was different from one which had been predicted by the HYSPLIT simulations. It should be noted that the mobile DOAS observations were organised in such a way that the data on the TrC of $NO_2$ for the location outside the city plume were collected first. There were two days of FTIR measurements without mobile DOAS observations due to technical issues. Our experience has shown that the HYSPLIT forecast was precise enough to ensure proper selection of FTIR locations on these days.

## 4 Methods and algorithms of the experimental data processing

### 4.1 FTIR and DOAS data processing

The dual-channel EM27/SUN spectrometer can measure TCs of $O_2$, $H_2O$, $CO_2$, $CH_4$ and CO (Gisi et al., 2012; Hase et al., 2016). The processing of the raw FTIR data (generation of spectra from raw interferograms and trace gas retrievals) is performed using the software tools provided by the COCCON (Frey et al., 2019; COCCON, 2019). The required software is source-open and freely available; the development of these tools has been supported by ESA. The interferograms recorded with FTS#80 and FTS#84 were the main input data. In the first processing step, spectra are generated from the recorded DC-coupled interferograms, including a DC correction (Keppel-Aleks et al., 2007) and quality filtering. In the second processing step, total column abundances (TCs) of the target species are derived from the spectra. For the retrievals of the total columns of $O_2$, $CO_2$, CO, $H_2O$, and $CH_4$, the spectral regions recommended by Frey et al. (2019) and Hase et al. (2016) were taken. We present these intervals in the respective order: $7765 - 8005$ cm$^{-1}$ (the main interfering gases are $H_2O$, HF, $CO_2$), $6173 - 6390$ cm$^{-1}$ (the main interfering gases are $H_2O$, HDO, $CH_4$), $4210 - 4320$ cm$^{-1}$ (the main interfering gases are $H_2O$, HDO, $CH_4$), $8353 - 8463$ cm$^{-1}$, and $5897 - 6145$ cm$^{-1}$ (the main interfering gases are $H_2O$, HDO, $CO_2$). The EM27/SUN spectrometer has low spectral resolution of 0.5 cm$^{-1}$. Therefore the TCs are derived from the FTIR spectra by scaling of a priori profiles of target gases (Frey et al., 2019). The required auxiliary data are the local ground pressure, the temperature profile and the a priori mixing ratio profiles of the gases. For ensuring consistency with the TCCON reference network in this regard, these atmospheric profiles were provided by TCCON. The ratio of the target gas TC to the retrieved $O_2$ TC which is suggested to be known and constant, gives us the column-averaged dry-air mole fraction ($X_{gas}$) of the target gas (Wunch et al., 2011; Frey et al., 2015):

$$Xgas = 0.2095 \frac{TCgas}{TC_{O2}} = \frac{TCgas}{TCdry\ air}, \tag{1}$$

where $X_{gas}$ - column-averaged dry-air mole fraction of the target gas (unit: dimensionless quantity), $TC_{gas}$ – total column of the target gas (unit: molec. m$^{-2}$), $TC_{O2}$ - total column of $O_2$ (unit: molec. m$^{-2}$), $TC_{dry\ air}$ – dry air total column (unit: molec. m$^{-2}$). Using Xgas helps to reduce the effect of various possible systematic errors (Wunch et al., 2011). To provide the compatibility of EM27/SUN measurements to WMO scale and for consistency reasons, the retrieval software used for processing the EM27/SUN spectra also performs a post-processing (Frey et al., 2015). Finally, we had at our disposal both the TCgas and Xgas for each day of measurements at each observational location.

For the interpretation of spectral UV-VIS measurements and the derivation of tropospheric $NO_2$ content, the well known DOAS method is used (Platt and Stutz, 2008). Basically, DOAS algorithm derives the $NO_2$ atmospheric content by fitting a

reference $NO_2$ absorption cross-section to the measured zenith scattered radiance. The effective or slant column density (SCD) of $NO_2$ is retrieved in the 425-485 nm fitting window. SCD is converted then to vertical column density (VCD) by means of so-called air mass factor, AMF (VCD=SCD/AMF), pre-calculated with a radiative transfer model (RTM). The spatiotemporal variations of stratospheric $NO_2$ are negligible compared to these in a polluted troposphere. Consequently, the variations of $NO_2$ vertical column observed in the data of our mobile DOAS measurements are related to $NO_2$ pollution in the boundary layer (below ~1.5 km). In general, such observations have been proved to be an efficient technique to derive the anthropogeinc $NO_x$ flux in many studies worldwide (see e.g., Johansson et al., 2008, Rivera et al., 2009, Johansson et al., 2009, Rivera et al., 2010, Ibrahim et al., 2010, Shaiganfar et al., 2011, Wang et al., 2012, Shaiganfar et al., 2015, Wu et al., 2017, Shaiganfar et al., 2017).

## 4.2 Side-by-side calibration of FTIR spectrometers

The target quantity of our observations is the small difference between two large values that are measured by different instruments of the same type. Therefore, a careful cross-calibration of the instruments is of primary importance for the considered experiment. Side-by-side calibrations of FTS#80 and FTS#84 were carried out during four days: 12 April, 26 April, 15 May, and 16 May, 2019. The instruments were installed at the observational site of St. Petersburg State University in Peterhof and operated simultaneously for the time period of clear sky weather which lasted from half an hour to several hours. The total number of spectra acquired during cross-calibrations was 604. They were collected during about 10 h of simultaneous measurements. The scatter plots showing cross-comparison of the data are given in Fig. 4. For all considered gases ($CO_2$, $CH_4$, CO), the results for column-averaged dry-air mole fractions (Xgas) delivered by two FTS are in a very good agreement. The determination coefficients for $CO_2$, $CH_4$ and CO are 0.9999(99), 0.9999(99), and 0.9999(89) respectively. The calibration factors obtained as a result of side-by-side comparison were used to convert $XCO_2$, $XCH_4$, and XCO measured by spectrometer #80 to the scale of spectrometer #84. The results of cross-calibration help to avoid an additional source of systematic error in the estimation of area fluxes. The RMS differences between time series of simultaneous measurements by FTS#80 and FTS#84 are equal to 0.10 ppm (0.025%) for $CO_2$, 0.59 ppb (0.032%) for $CH_4$, and 0.38 ppb (0.38 %) for CO.

The scaled results of the side-by-side measurements of $XCO_2$, $XCH_4$, and XCO by FTS#80 and FTS#84 on 12 April 2019 at the St. Petersburg observational site are presented in Fig. 5. The individual results and 15 min running average data are shown. We used the side-by-side measurements for estimating the optimal averaging period for the Xgas data. Averaging is the necessary prerequisite for using these data for the evaluation of emission and for comparison with the results of modelling. It should be emphasized that the data sampling for other input parameters is varying considerably. In order that all datasets are consistent, the optimal sampling intervals were determined. For the FTIR measurements, the averaging

interval has been selected in such a way that short term variations of measured quantities can be detected. As an example, we point at three local maxima of $XCH_4$ and $XCO$ during the time period of 13:00-15:00. One can see that these maxima with the "half width" of about 15-20 min and with the amplitudes of ~0.5 ppbv and of 0.1 ppbv for $XCH_4$ and $XCO$ respectively are nicely covered as well as the increase of the greenhouse gases around noon, so the chosen value of averaging interval of 15 min seems reasonable. The chosen averaging interval of 15 min is in good agreement with the estimation of the opimal integration time (10 min) obtained as a result of the Allan analysis implemented by Chen et al. (2016). Chen et al. (2016) applied this approach for the differential measurements of $XCO_2$, $XCH_4$ performed by three EM27/SUN spectrometers within urban areas.

## 4.3 Mass balance approach for area flux estimation

The estimation of the area fluxes $F$ was obtained on the basis of a mass balance approach implemented in the form of a one-box model. Box models are a widely used technique for the evaluation of urban and other emission fluxes (Hanna et al., 1982; Reid and Steyn, 1997; Arya, 1999; Zinchenko et al., 2002; Zimnoch et al., 2010; Strong et al., 2011; Hiller et al., 2014a; Chen et al., 2016; Makarova et al., 2018). In our case the following equation for the calculation of area flux was used:

$$F_j(t_k) = \frac{\Delta_{TC}(t_i) \cdot V_j(t_i)}{L_j(t_i)} \cdot k \,, \tag{2}$$

where $F$ (unit: t km$^{-2}$ yr$^{-1}$) is the area flux, $t_i$ denotes the day of a single field experiment in the frame of the observational campaign. It should be emphasized that we used the steady-state approximation for all involved processes within the duration of a single field experiment, so $\Delta_{TC}$ (unit: molec. m$^{-2}$) is the mean TC difference between downwind ($TC_d$) and upwind ($TC_u$) observations $\Delta_{TC} = TC_d - TC_u$, $V$ (unit: m sec$^{-1}$) is the mean wind speed, and $L$ (unit: m) is the mean length of a path of an air parcel which goes through the urban territory of St. Petersburg agglomeration. The $k$ coefficient converts the value of area flux from (unit: molec. m$^{-2}$ sec$^{-1}$ ) to (unit: t km$^{-2}$ yr$^{-1}$):

$$k = \frac{m_{gas} \cdot 31536 \cdot 10^6}{N_A} \,, \tag{3}$$

where $m_{gas}$ is the molecular mass of the target gas (unit: kg mol$^{-1}$), $N_A$ – Avogadro constant (unit: mol$^{-1}$), $31536 \cdot 10^6$ - the coefficient that converts the value of area flux from (unit: kg m$^{-2}$ sec$^{-1}$) to (unit: t km$^{-2}$ yr$^{-1}$). The data for the wind speed and the wind direction were taken from different sources of meteorological information (see section 4.3), and these sources are identified as $j$ in Eq. 2. So, as a result, we obtained the set of values of $F(t)$ for each of the meteorological data sources and for each day of field measurements. We note that below we will use the units t km$^{-2}$ yr$^{-1}$ for the values of $F(t)$.

## 4.4 Wind field data

Obviously, reliable wind field information is an important prerequisite to get an accurate estimate of the target emissions from the data of remote spectroscopic measurements. For instance, it has been noted by Ionov and Poberovskii (2015), that the uncertainty of the surface wind direction is the main contributor to the total error of $NO_x$ emission by the megacity of St. Petersburg, estimated from circular DOAS measurements. It was also found that the direction of the surface wind acquired by ground-based meteorological observations often does not match the results of modelling of the pollution plume and the

results of the $NO_2$ mobile measurements (Ionov and Poberovskii, 2017). Apparently, the routine wind observations in the city are subject to significant local perturbations due to unavoidable interactions of the wind flow and the adjacent city buildings. It should be emphasised that the HYSPLIT simulations of the fields of tropospheric $NO_2$ demonstrate reasonable agreement with the plume dispersion observed by the circular mobile observations (Ionov and Poberovskii, 2017; Ionov and Poberovskii, 2019). The latter is also true for plume simulations, presented in the current study in Fig. 3. However, one can

easily notice inconsistencies between the dominant directions of plume movement and the surface winds as specified in Table A2 (see Appendix A): e.g. days March 21, March 27, April 1 and April 24, when the city plume was moving southeast but the surface wind was west-southwest (see Fig. 3). In order to get more accurate wind information, we have considered additional sources of wind data:

- in situ measurements of Vaisala weather transmitter WXT520 with an ultrasonic wind sensor, installed locally on the

roof of the building of the Institute of Physics of SPbU (~60 m a.s.l, 59.88°N, 29.83°E, point A1 in Fig. 2); hereafter mentioned as "LOCAL";

- the data of Global Data Assimilation System (GDAS) from NCEP GFS model, which is similar to the one used to initialize the HYSPLIT dispersion calculations as specified in Section 3; hereafter mentioned as "GDAS";

- the wind speed and direction data retrieved from the backward trajectory calculations of HYSPLIT at the location of

downwind FTIR observation; hereafter mentioned as "HYSPLIT".

We selected HYSPLIT as one of the sources of the wind data since HYSPLIT is a widely used modelling system for the simulation of air parcel trajectories and the dispersion processes in the atmosphere which was tested in a lot of studies (HYSPLIT publications can be found using the following links: https://www.arl.noaa.gov/hysplit/hysplit-publications-meteorological-data-information/). Stein et al. (2007) noted that *Grid models are the best-suited tools to handle the regional*

*features of these chemicals. However, these models are not designed to resolve pollutant concentrations on local scales. Moreover, for many species of interest, having reaction time scales that are longer than the travel time across an urban area, chemical reactions can be ignored in describing local dispersion from strong individual sources making Lagrangian and plume-dispersion models practical.* Stein et al. (2007) classify HYSPLIT as a local model which provides *the more spatially resolved concentrations due to local emission sources*. Therefore, for modelling of the evolution of the

St.Petersburg plume we used the HYSPLIT model as a tool which perfectly fits the scale of considered atmospheric processes. This was also the reason for using HYSPLIT as the source of the wind data.

Both "GDAS" and "HYSPLIT" wind data are taken at the altitude level that approximately corresponds to the middle of the daytime boundary layer height. An average wind is calculated for the time period of FTIR observations. Resulting wind speeds and directions from the three different data sources are given in Table A3 (see Appendix A). As expected, wind
speeds at elevated altitude levels from GDAS and HYSPLIT are much higher than the surface wind speeds (see Appendix A, Table A2). On some days, e.g. April 6 and April 18, in situ wind directions ("LOCAL") differ considerably from "GDAS" and "HYSPLIT", although the latter two are consistent with each other. Note that compared to surface, the elevated wind directions better reproduce the city plume movement – e.g. northwest and west-northwest directions on days March 21, March 27, April 1 (see Fig. 3) instead of west-southwest at the surface (see Appendix A, Table A2).

**4.5 Air parcel path length**

The determination of the air parcel path length $L$ (Eq. 2) is a sophisticated task due to the fact that the application of a box model suggests that the pollutants are well mixed in the entire air box volume, but it is not true, especially for megacities with complex structure of the urban terrain and distribution of emission sources. Thus, different approaches have been tested to calculate $L$:
- Simplified box model setup with a constant path length $L_j(t_i)=L=\mathrm{const}$ for each day of field observations. The box is designed to represent the major part of high density residential and industrial area of the St. Petersburg agglomeration, so that respective $L$ is derived from the value of that area. Since the locations of our field observations are mostly placed on the outer side of the ring road, this road was set to be a boundary for the target emission area. Accordingly, given that the land area inside the ring is equal to 706 km$^2$, we get an estimate of $L=\sqrt{706}\approx27$ km. Hereafter the results of data
interpretation by means of this approach are indicated by "$L_{const}$".
- The variable effective path is calculated using the actual wind direction and the land use pattern on the route of the linear air trajectory. Only those sections of path are being taken into account that cross the area of supposed anthropogenic emission. The input wind directions are those mentioned above in Table A3 (see Appendix A), and the resulting path length calculations hereafter are indicated as "$L_{LOCAL}$ ", "$L_{GDAS}$ " and "$L_{HYSPLIT}$ ". The use of the effective path in Eq. 2
takes into account to some extent the inhomogeneity of the anthropogenic emissions in the megacity.

For the purpose of effective paths calculation, a special gridded model of land use coverage has been constructed on the basis of the visual classification of publicly available map (https://yandex.ru/maps/2/saint-petersburg/?ll=30.163886%2C59.911377&z=11, access date 28 January 2020) that covers the St. Petersburg agglomeration with its surroundings (see Fig. 6). The spatial domain of the model covers 76 km in south-north direction and 128 km in east-

west direction (59.60-60.29°N, 29.05-31.33°E). It has been assumed that there are no significant emission sources outside this domain. The model resolution (grid size) is 25 m × 25 m. The following major land use classes are considered: residential buildings/industrial areas, roads/highways, water bodies, parks/forests/fields, and swamps/wetlands. In Fig. 6 these land use classes are shown in different colours: blue for the water bodies, grey for the residential buildings/industrial areas, green for the parks and forests. Effective path length is calculated as a sum of elementary paths through the urbanized grid pixels which contain residential buildings, industrial areas, and roads/highways. Pixels containing water bodies, swamps, and parks are excluded from the variable path calculations. Similar approach was implemented by Hase et al. (2015). The total urbanized area of the St.Petersburg agglomeration according to the developed land use classification occupies the area of 984 km$^2$ while the official area of the entire St.Petersburg is of 1439 km$^2$. The target gases can orginate from different emission source categories, i.e. $CH_4$ could partly come from the waterways (sewers and water canals), wetlands and pipelines rather than mobile and point combustion sources which are relevant for CO, $CO_2$ and $NO_2$. The EMME-2019 was carried out during March-April when the water bodies and earth surface were fully or partly covered by ice and snow (see Appendix A, Fig. A1), and soils were still frozen. Therefore we suggest that $CH_4$ emission from the excluded pixels (water bodies, swamps, parks, and forests) was negligible in comparison to other anthropogenic sources (landfills, pipelines and etc.) which are distributed over the urbanized pixels.

To minimize errors that may occur due to the land use misclassification and to take into account the airflow spatial extension, the 10 km wide band of 11 equidistant and parallel paths is analyzed and an average path length is calculated. Finally, the difference between the "polluted" path (backward from the downwind location) and "clean" path (backward from the upwind location) provides an estimate of the effective path $L$. Fig. 6 presents an example of linear backward paths for the days of FTIR observations with the major land use classes shown by different colours.

**4.6 Case study: two examples**

In order to illustrate the interpretation of experimental data and describe the main error sources of final results, we consider two days of field measurements. The first one, April 4, seems to be the most successful in terms of observational conditions, functioning of the equipment, data quality and clarity of the interpretation. It is characterised by stable weather conditions with a moderate south-southwest wind, similarly identified by different wind data sources – from the surface (see Appendix A,Table A2) to higher altitude levels (see Appendix A,Table A3). The simulated city plume picture demonstrates a jet-like flow of air mass on that day, with almost perfect location of both FTS, upwind and downwind almost on one line (see Fig. 3). Besides, according to the model simulation for April 4, the upwind FTS was located in the clean area, while the downwind one was installed very close to the plume jet. Another example is April 25, when both FTS locations appeared to be inside the polluted area. This happened due to the specific weather conditions that contribute to the accumulation of air

pollutants in the boundary layer: calm night before and light winds of 1 m s$^{-1}$ in the day time (see Appendix A Table A2 and A3). Moreover, the wind direction on April 25 at the surface (south-southwest, Table A2) is very different from that in the middle of the boundary layer (east and east-northeast, Table A3).

     According to the analysis of the air samples collected in air bags, the surface air on April 25 was extremely polluted. The downwind $NO_2$ concentration was found to be 138 μg m$^{-3}$, while it was varying within the range of 12-74 μg m$^{-3}$ during
the other days of field observations. Another indication of heavy anthropogenic pollution comes from the data of our mobile DOAS measurements: the maximum of $NO_2$ TrC registered along the circular route was $92 \cdot 10^{15}$ molecules cm$^{-2}$ on April 25, while it was in the range of $15\text{-}58 \cdot 10^{15}$ molecules cm$^{-2}$ on the other days of field observations. According to the data of municipal air quality monitoring, the daily average concentration of the particulated matter (PM10) was very high and exceeded 60 μg m$^{-3}$ (http://www.infoeco.ru/, last access 4 March 2020). High pollution event was registered also by the
CIMEL sun photometer installed at St. Petersburg State University (point A1, Fig. 2) within the AERONET international programme (Volkova et al., 2018): the daily averaged value of aerosol optical thickness (AOT) at 500 nm was found to be 0.40 on April 25 which is considerably higher than its long term average value (0.12 for the period of 2013-2019); similar increase of AOT was registered by the satellite measurements of the MODIS satellite instrument over St. Petersburg on that day.

The TC data of $CO_2$ measurements on April 4 and April 25, with a 15-min running averages, are presented in Fig. 7. Compared to April 4, the TC of $CO_2$ on April 25 demonstrates higher levels and variation, both at upwind and downwind locations. Although the downwind TC is generally below the upwind level, as expected, the upwind TC starts to exceed downwind level at the end of FTS observations on April 25. Accordingly, while the "downwind-upwind" difference is relatively stable within the range of $2\text{-}4 \cdot 10^{19}$ molecules cm$^{-2}$ on April 4, it reaches $10 \cdot 10^{19}$ molecules cm$^{-2}$ at 12:00 on
April 25, but becomes zero and then negative (up to $-1 \cdot 10^{19}$ molecules cm$^{-2}$) after 14:30. In order to explain this behaviour, a special run of HYSPLIT dispersion model was performed, with an output of $CO_2$ TC within a boundary layer every 15 minutes, at both FTS locations, upwind and downwind (see Fig. 7). As the first approximation, the $CO_2$ emission sources were assumed to be located similar to the $NO_x$ emission sources but scaled to match the level of our FTS measurements. These calculations qualitatively reproduce the time series of the $CO_2$ measurements and the different character of the results
of field experiments on April 4 and April 25. Moreover, we can suggest that the origin of high $CO_2$ TC values observed at the upwind FTS location on April 25 was the thermal power station located about 5 km towards north from the upwind point (see Fig. 6). When the emission by the thermal power station is turned off in the HYSPLIT calculation, the $CO_2$ TC drops down to the level of upwind FTS measurements on April 4 (see Fig. 7b, blue dashed line).

The time series of Xgas for $CO_2$, CO and $CH_4$ obtained from the data of FTS measurements on April 4 and April 25 are shown in Fig. 8. Since the Xgas variability at clean location (upwind) is usually much smaller as compared to a polluted location, it is possible to use time extrapolation of measured data for the periods with data gaps. Fig. 9 demonstrates the difference between TC for each of three gases measured by upwind and downwind FTS on April 4 and April 25; the extrapolated data are specially marked. Fig. 9 also shows the wind speed and wind direction for the time period of FTS observations by the "LOCAL" weather station (see section 4.3).

## 5 Results and discussion

### 5.1 Overview of obtained results

The campaign consisted of 11 days of field measurements. On 30 April the clouds (altocumulus translucidus) started to develop quickly during the field experiment (see Appendix A Table A1 and Fig. A1). On 18 April the upwind FTS location was close to the thermal power station. Owing to the prevailing north-northeast wind (see Appendix A Table A3), the upwind FTS location appeared to be polluted on 18 April (see Fig. 3). Consequently, 18 April and 30 April were excluded from final analysis, and the evaluation of the target fluxes ($F$) of the investigated gases was limited to remaining 9 days of campaign. For these 9 days the cross-correlations (Pearson's correlation coefficient $r$) between $\Delta_{TC}$ values obtained for the pairs $CO/CO_2$ and $CH_4/CO_2$ were calculated: $r_{CO/CO2} = (0.88 \pm 0.02)$; $r_{CH4/CO2} = (0.82 \pm 0.03)$. The high correlation is the evidence of the fact that the measurements in most cases were conducted inside the plume coming from a regional/mesoscale relatively compact powerful source of emission. We can attribute this source to the centre of St. Petersburg.

To further consolidate our flux estimates, some additional restrictions were imposed on the experimental data, which resulted in keeping only 4 days out of 9: March 21, March 27, April 3 and April 4. The first requirement was the wind field stability. The analysis of the wind field stability during each day was carried out using the GDAS and HYSPLIT meteorological data, as well as local meteorological observations. The second criterion was the homogeneity of the megacity pollution plume. It was estimated on the basis of the analysis of the daily variability of enhancement ratios $EnhR = \Delta_{TC,gas1}/\Delta_{TC,gas2}$. The $EnhR$ values for the following pairs were considered: $CO/CO_2$ and $CH_4/CO_2$. For selected days, the upper limit of the daily relative variability of $EnhR$ was set as 30%.

As it has been described above, there were several different scenarios of the $F$ calculations in which different sources of meteorological information (LOCAL, GDAS, and HYSPLIT) and different methods of the air parcel path calculations were used. The comparison of the obtained results has shown that the minimum variability of $F$ is observed when the HYSPLIT meteorological data are combined with the variable effective path $L$ (see section 4.5). When selecting the results for final analysis, we suggest that the application of the criterion of minimal variability is a good choice because in this case

the corresponding estimates of area flux are more reliable. This statement can be confirmed in particular by comparison of the $CO_2$ fluxes obtained for the 9-day and 4-day sets (Table 1, columns 2 and 3). For the 4-day set, the variability is considerably lower (12 vs. 28 kt km$^{-2}$ yr$^{-1}$), and we should reiterate, that these 4 days were the days with the most favourable observational conditions during the observational campaign. So, we do not present the results of all scenarios, and show in Table 1 (columns 2 and 3) the values obtained for the combination of HYSPLIT meteorological data with the variable effective path. As a supplementary information, in the Appendix B we placed Table B1 which contains the values of area fluxes for $CO_2$, $CH_4$, CO, and $NO_x$ obtained using constant path length approach.

If we compare the flux values obtained for the 4-day and 9-day sets, we see that the fluxes for $CO_2$ are the same, but the fluxes for $CH_4$ and CO are different (Table 1, columns 2 and 3). The fluxes estimated for the selected 4 days appeared to be 1.3 times higher than corresponding values obtained for all 9 days of field observations. The uncertainty of the obtained flux values for the 4-day subset decreased for $CO_2$ and $CH_4$. We stress that during these selected 4 days not only the specific meteorological conditions corresponded in the best way to the assumptions of the box model, but also the locations of the observational points were nearly perfect.

The summary of the EMME-2019 results and the comparison with the flux estimates for St. Petersburg based on in situ measurements, as well as independent literature data, are presented in Table 1 for $CO_2$, $CH_4$, CO and $NO_x$ (the latter were derived from mobile DOAS measurements of tropospheric $NO_2$ in the vicinity of upwind and downwind FTIR observations). Prior to analysis of the results, a short overview of the error and uncertainty analysis should be presented. The random uncertainty of mean $F$ values of $CO_2$, $CH_4$, CO, and $NO_x$ indicated in Table 1 was calculated as STD of daily means of area fluxes. This uncertainty includes two components. The first component is the natural flux variability and the second component comprises the random measurement errors and the errors introduced by approximations and simplifications of the model approach which was used. It should be specially emphasised that these two components cannot be identified separately. Therefore, below we will use the terms "variability" or "uncertainty" keeping in mind that these terms denote natural variations, measurement errors and model errors together. The relative random uncertainty of $F$ for one specific day of measurements (daily uncertainty) can be estimated using the following expression:

$$\delta F = \delta V + \delta L + \delta \Delta_{TC} \tag{2}$$

where $\delta V$ is the relative variation of the wind speed over a day estimated using HYSPLIT meteorological data, $\delta L$ is the relative uncertainty of the air parcel path length, and $\delta \Delta_{TC}$ is the relative daily variation of $\Delta_{TC}$. The $\delta F$ values calculated in this way can be considered as an upper limit of the $F$ uncertainty. The average values of $\delta L$, $\delta V$ and $\delta \Delta_{TC}$ estimated for 9(4) days of the city campaign are as follows: $\delta L = 23(24)\%$, $\delta V = 23(13)\%$, $\Delta_{TC}(CO_2) = 33(28)\%$, $\Delta_{TC}(CH_4) = 50(22)\%$ and

$\Delta_{TC}(CO) = 42(28)\%$. Finally, the average values of relative daily uncertainty of area fluxes are equal to $\delta F_{CO2} = 79(65)\%$, $\delta F_{CH4} = 96(59)\%$ and $\delta F_{CO} = 88(65)\%$. As an example, daily mean values of $CO_2$ area flux obtained during the city campaign are presented in Fig.12 where the "error bars" are the random uncertainties of $F$ values derived from corresponding relative mean uncertainties for 9(4)-day sets.

To evaluate systematic error of the area flux ($\delta F_{sys}$) we should first estimate the systematic errors $\delta L_{sys}$, $\delta V_{sys}$ and $\delta \Delta TC_{sys}$ of corresponding parameters $L$, $V$ and $\Delta TC$ in Eq.2. In contrast to $\delta L_{sys}$ and $\delta V_{sys}$, the contribution of systematic component of $\delta \Delta TC_{sys}$ into $\delta F_{sys}$ is negligible. This is due to the high accuracy of the COCCON observations of gas columns which are calibrated against WMO scale. In Eq. 2 we use an assumption that an air parcel moves along a straight line but obviously this is not true. For the whole ensemble of HYSPLIT trajectories simulated for all days of the city campaign we calculated the maximum relative difference between the true lengths of HYSPLIT trajectories and our straight line approximations of $L$. This value equals to ~4% which is considered as an estimation of the relative systematic error $\delta L_{sys}$. According to the information on wind speed (see Appendix A, Table A3) observed during the field campaign, the mean relative difference between HYSPLIT and GDAS data on wind speed is of 14±22%. Hence, the estimation of the systematic error of area flux $\delta F_{sys}$ due to the systematic errors of all parameters in Eq.2 gives the value 18%.

## 5.2 Estimation of the CH4 emissions by means of in situ measurements of its mixing ratio

The fourth column of Table 1 contains the estimations of $F$ for the territory of St. Petersburg, which were made on the basis of the joint analysis of the $CH_4$ local concentrations monitored in the ambient air during March-April 2013 and April 2019 at the SPbU atmospheric monitoring station (point A1) (Makarova et al., 2018) and Voeikovo station (59.95°N, 30.70°E, 72 m above sea level) of the Voeikov Main Geophysical Observatory (MGO) (Zinchenko, 2002). The $CH_4$ measurements are carried out by MGO in accordance with WMO recommendations for GAW stations (WMO, 2009; WMO, 2014). The high quality of the data obtained by MGO is confirmed by the results of WMO/IAEA Round Robin Comparison Experiment 2014-2015 (https://www.esrl.noaa.gov/gmd/ccgg/wmorr/wmorr_results.php, last access 3 March, 2020). The data of Voeikovo station together with 17 other European stations were used to estimate European methane emissions in the framework of the InGOS project (Bergamaschi et.al., 2018). The measurements of these stations have been rigorously quality controlled (Lopez et al., 2015; Schmidt et al., 2014). The Voeikovo measurements are calibrated against the NOAA-2004 standard scale (which is equivalent to the World Meteorological Organization Global Atmosphere Watch WMO-CH4-X2004 $CH_4$ mole fraction scale) (Dlugokencky et al., 2005). The comparability of the SPbU and Voeikovo station data was ensured by calibrating the SPbU equipment against the working standard prepared by MGO.

Determination of the $CH_4$ fluxes is possible due to the beneficial location of the observational stations of SPbU and MGO - on the western and eastern sides of the megacity. For the wind directions of 75-85° and 255-265°, the air mass on the

way from one station to another passes through the centre of St. Petersburg. It should be emphasised that only the time periods with the wind speed of at least 2.5 m s$^{-1}$ were considered. Using the difference in the $CH_4$ concentrations obtained at the monitoring stations, it is possible to estimate the $CH_4$ flux for the central part of the St. Petersburg agglomeration on the basis of a simple box model similar to that used in the present work. It was assumed that all contaminations emitted by St. Petersburg into the atmosphere stay within the boundary layer. The calculation of the variable effective path $L$ between these two monitoring stations gives $(21 \pm 7)$ km. The HYSPLIT backward trajectory outputs were used as a source of meteorological data (wind field, boundary layer height data). Finally, the $F$ values for $CO_2$ and CO were estimated using the obtained average $CH_4$ flux $(120\pm80$ t km$^{-2}$ yr$^{-1})$ and average $EnhR$ values derived from the in situ measurements of the $CO_2$, $CH_4$, and CO concentrations at SPbU atmospheric monitoring station (point A1) in 2013-2019 (Table 2, the third column). The flux values for $CO_2$ and CO evaluated in this way are 2-3 times lower than the corresponding results of EMME-2019. First, we should emphasize that in-situ measurements are more sensitive to very local effects and therefore less representative if compared to column observations. And second, this difference can be partially explained by the presence on the territory of St. Petersburg of a significant number of elevated stationary sources of $CO_2$ and CO – industrial and power/heat plant chimneys (chimneys of the power plant stations can have a height of ~200 m), which emit products of combustion and oxidation of various types of fossil fuels. The effect of elevated sources on gas concentrations measured at the surface layer is often minimal, but this impact can be considerable for total/tropospheric columns and can be detected using remote sensing techniques such as those used during the Berlin campaign (Hase, et al., 2015) and EMME-2019. We present more discussion on this topic in Appendix C. In order to detect the presence of the elevated sources, the air sampling using kite launches was performed during EMME-2019. The air sampling by kite launching technique was possible only twice when suitable wind speed conditions occurred and there was enough free space for launching. The results of comparison of the gas concentrations in air samples collected at the surface and elevated levels on 24 April 2019 and on 25 April 2019 at the locations of FTS measurements inside the city plume are presented in Table 3. In most cases the concentrations of considered gases at the elevated level are lower if compared to the surface level. There were only two cases with the concentration enhancement in the air samples collected by kite: for $CH_4$ on 24 April and for $CO_2$ on 25 April, however these enhancements were negligibly small (1 ppbv for $CH_4$ and 1 ppmv for $CO_2$).  So, one can come to the conclusion that these two kite launches revealed no elevated pollution plumes.

### 5.3 Comparison with inventories

Official reports on the environmental conditions of St. Petersburg (Serebritsky, 2018, 2019) contain information on the annual emissions of $CO_2$, $CH_4$, CO and $NO_x$ for the entire territory of the metropolis. For comparison with our flux estimates, these total rates were divided by the urbanized area of St. Petersburg (984 km$^2$, see section 4.5). The best agreement of the results of the EMME-2019 campaign with the official emission inventory was obtained for $NO_x$ and CO.

For $NO_x$, the results of the field campaign and the official emission inventory demonstrated close values: 66 t km$^{-2}$ yr$^{-1}$ and 69 t km$^{-2}$ yr$^{-1}$. The average CO flux for the territory of St. Petersburg, according to official data, is 410 t km$^{-2}$ yr$^{-1}$, which is higher in comparison with the values obtained in the current work (251-333 t km$^{-2}$ yr$^{-1}$). At the same time, a significant differences in the $F$ estimates for $CH_4$ and $CO_2$ were obtained: the official data are by 5-7 and 3 times lower than the corresponding values obtained during field observations in March-April 2019. Hiller et al. (2014a) showed that the application of the boundary layer budget approach in the form of a box model could give the $CH_4$ area fluxes of about 1.5-2 times higher in comparison with corresponding values estimated by eddy covariance technique and 2.5-6 times higher than $F$ derived from the emission inventory data.

The results of independent studies of anthropogenic emissions reported in the scientific literature show that the estimates of the $CO_2$, $CH_4$, CO, and $NO_x$ fluxes can vary in a very wide range depending on season, meteorological situation, location of observation points, measurement technique, and used approach for estimation of emission (Vaughan et al., 2016; Hiller et al., 2014a; and also see the references indicated in Table 1). The $CO_2$ flux for the St. Petersburg agglomeration obtained in this paper is approximately three times higher than those for London and Berlin and ~7 times higher than for Tokyo and Mexico City (see Table 1). We would like to note that when comparing the results of different observational campaigns one should pay attention to the seasonal features of emissions. For example, the Berlin campaign took place in early summer when space heating was off. The EMME-2019 campaign in St. Petersburg was carried out in March-April. The space heating in St. Petersburg is mainly organised as the system of district heating which is running in the winter mode during this period. The district heating in St. Petersburg is usually turned off in the beginning of May. For $CH_4$, the emission intensity is about 2-3 times higher than the results for London. The CO fluxes for megacities, according to published data, can demonstrate a wide range of values, for example, varying from 106 t km$^{-2}$ yr$^{-1}$ (London) to 1520 t km$^{-2}$ yr$^{-1}$ (Mexico City). This range covers our estimates for St. Petersburg: ~251-333 t km$^{-2}$ yr$^{-1}$.

One of the most important characteristics of the air pollution source is the emission ratio $ER_{gas1/gas2}$:

$$ER_{gas1/gas2}=F_{gas1} M_{gas2}/(F_{gas2} M_{gas1}), \tag{3}$$

where $F_{gas}$ is the gas flux, $M_{gas}$ is the molecular weight of gas. For gases, such as $CO_2$, $CH_4$, and CO, whose lifetime in the troposphere is significantly longer than the duration of field measurements (several hours), the following equality is valid: $ER = EnhR$. The $ER$ values obtained from the results of the EMME-2019 campaign and in situ measurements at the SPbU atmospheric monitoring station (point A1) in 2013-2019, as well as $ER$ calculated for the official emission inventory and the $ER$ taken from literature are presented in Table 2. The emission ratios for St. Petersburg obtained as a result of the EMME-2019 campaign and of the in situ monitoring of $CH_4$ at the observational stations located near St. Petersburg have similar values, which are in good agreement with the information on $ER$ for the world's largest cities reported in literature. For the official emission inventory, the $ER$ values for $CO/CO_2$ and $CH_4/CO_2$ correspond to the upper and lower limits of the

given literature data, respectively. Thus, the relative contributions of $CO_2$, $CH_4$ and $CO$ to the total emissions of the St. Petersburg agglomeration are very similar to the corresponding values for the world megacities.

## 5.4 Identification of problems

When studying the application of the remote sensing instruments to the problem of the air pollution meteorology, Beran and Hall (1974) noted:

> "Every urban region is a unique entity and the correct location and sensor distribution for one city may be totally unacceptable for another. Certain features are, however, common to all and can be used to generate a hypothetical city."

Such hypothetical city usually contains industrial region and line sources of emission in the form of highways. Beran and Hall (1974) also made the following important remark:

> "Terrain features are another important influence on urban meteorology, many times controlling the local flow which advects or concentrates effluent in a given region. For example, a river valley is a natural place for cold air drainage, while a coast line produces local land and sea breeze circulation, alternately cleansing a region and concentrating pollution at the sea breeze front."

All these mentioned terrain features are present on the territory of the St. Petersburg agglomeration. St. Petersburg is located at the estuary of the Neva River which flows in the Gulf of Finland. The territory of St. Petersburg occupies northern, eastern and southern coastlines of the Gulf of Finland (Fig. 2). About 40 km to the north-east from the centre of St. Petersburg, the southern coastline of the Ladoga Lake is located. The Ladoga Lake is the largest lake in Europe. All these facts define the weather and climate in St. Petersburg. The complex terrain of St. Petersburg agglomeration requires special attention due to its influence on the air pollution meteorology.

The number of sunny days in St. Petersburg is not large. We tried to use every clear-sky day. But the weather in St. Petersburg is unstable and in several cases the forecast for clear-sky was wrong. When it happened the field measurements which were already prepared for start were cancelled. On the other hand, there were clear-sky periods which were not forecasted. In some of such cases we managed to quickly organise and perform the field observations. As a result of unstable weather, the experiment appeared to be time consuming and interfering with other ongoing activities.

The measurement locations for two EM27/SUN instruments were appointed about 12 hours prior to the day of field campaign on the basis of the HYSPLIT forecast of the city plume dispersion. Moreover, during the field measurements there was a possibility to correct the locations on the basis of the $NO_2$ tropospheric column mobile measurements along the ringroad. Nevertheless, we could not implement the perfect setup of the experiment when both measurement locations of EM27/SUN were strictly on the straight line parallel to the wind direction. The problem arises from the sparsely distributed

sites suitable for installing the equipment and making observations. Also, we were limited in time since the travel time to the initial destination points was about 1 h and more. Changing of position is also time consuming process which includes the equipment loading, unloading and the travel time itself. The air sampling at different elevations by means of kite launching technique was possible only twice when the wind speed was suitable and there was enough free space for launching.

There is a certain problem relevant to the meteorological data obtained from different sources. First of all, a kind of ambiguity exists in selecting the optimal data source. The reason for that is different spatial and temporal distribution of data provided by different sources. Second, the data can be updated, for example we noted the updates of GDAS data sets which contained the considerable alteration of information.

## 6 Summary and outlook

We presented the description and the first results of the Emission Monitoring Mobile Experiment (EMME-2019) which was carried out in March-April 2019 in St. Petersburg, Russia. The main goal of this activity was the evaluation of emissions of $CO_2$, $CH_4$, CO and $NO_x$ for the megacity with the population of 5 million. The field campaign was performed in the area of the St. Petersburg agglomeration by joint efforts of St. Petersburg State University (Russia), Karlsruhe Institute of Technology (Germany) and the University of Bremen (Germany). The principal feature of EMME is its integrated character: several different instruments are used, and besides, the planning of the field experiment and data processing are performed with the help of numerical modelling of the transport of the megacity pollution plume. The concept of EMME is based on remote measurements of the total column amount of $CO_2$, $CH_4$ and CO from two mobile platforms located inside and outside the city plume combined with the mobile circular measurements of tropospheric column amount of $NO_2$ from the third non-stop moving platform, the latter measurements are used for the real-time control of the megacity plume evolution.

The results demonstrate that a combination of daytime synchronous upwind and downwind FTIR observations by two well-calibrated ground-based EM27/SUN FTIR spectrometers allow the reliable detection of $XCO_2$, $XCH_4$ and XCO enhancements due to urban emissions in the area of our study. The origin and temporal evolution of these enhancements were confirmed by simultaneous mobile DOAS measurements of tropospheric $NO_2$ around the city, the upwind and downwind in situ air sampling (with further analysis of $CO_2$, $CH_4$, CO and $NO_x$ concentrations), and by the simulations of urban pollution transport with the help of the HYSPLIT dispersion model calculations.

The collected data of our field campaign, supplemented with the precise in situ measurements of the $CH_4$ local concentrations at two sites in the suburbs of the city, allowed to get an estimates of the emission fluxes of greenhouse ($CO_2$, $CH_4$) and reactive (CO, $NO_x$) gases by the megacity of St. Petersburg. Resulting values reveal considerably higher emissions

of $CH_4$ ($135\pm 68$ t km$^{-2}$ yr$^{-1}$) and $CO_2$ ($89\pm28$ kt km$^{-2}$ yr$^{-1}$) if compared to the existing inventories, while our estimates of the CO emission ($251\pm104$ t km$^{-2}$ yr$^{-1}$) and $NO_x$ emission ($66\pm28$ t km$^{-2}$ yr$^{-1}$) are in agreement with the inventories.

The terrain of the St. Petersburg agglomeration is complex. It comprises the Neva river estuary and the coastline of the Gulf of Finland which influence the urban meteorology. Besides, multiple emission sources of different types and origin are inhomogeneously distributed over the main city and the suburbs. In the present study we used a simple box model approach for the derivation of the area fluxes of $CO_2$, $CH_4$, CO, and $NO_x$. Obviously, the application of more sophisticated models in combination with the detailed information on the emission inventory for the territory of St. Petersburg seems

promising for the continuation of the present study.

**Data availability**

The datasets containing the EM27/SUN measurements during EMME-2019 can be provided upon request; please contact Maria Makarova (m.makarova@spbu.ru) and Frank Hase (Frank.Hase@kit.edu).

**Author contributions**

MVM, FH, TB, and DVI conceived the study together. MVM, DVI, YAV, CA, VSK, SCF, MF, TW, AnVP, KAV, NAZ, YMT, EYB, SIO, BKM, AlVP, EFM and HKhI contributed greatly to the experimental part of the study. SCF, CA, and MVM were in charge of processing FTIR spectrometer data. DVI was in charge of numerical modelling of gas plumes and of conducting mobile DOAS measurements. Together MVM, FH, TB, DVI, SCF, CA, VSK, NNP, and VMI analysed and interpreted the results. MVM, VSK, DVI, and SCF prepared the original draft of the manuscript with contributions from FH,

TB, CA, MF, YMT, NNP, and VMI. MVM, FH, TB, DVI, VSK, MF, AlVP, and YMT reviewed and edited the manuscript.

**Competing interests**

The authors declare that they have no conflict of interest.

**Acknowledgements**

Two portable FTIR spectrometers EM27/SUN were provided to St. Petersburg State University, Russia, by the owner -

Karlsruhe Institute of Technology, Germany, in compliance with the conditions of temporary importation in the frame of the VERIFY project. The procedure of temporary importation of the instruments to Russian Federation was conducted by the University of Bremen, Germany. Ancillary experimental data were acquired using the scientific equipment of "Geomodel"

research centre of St. Petersburg State University. The authors gratefully acknowledge the NOAA ARL for the provision of the HYSPLIT transport and dispersion model and/or READY website (http://www.ready.noaa.gov, last access 2 March
2020) used in this publication. The operation of the CIMEL sun photometer and Los Gatos Research GGA 24r-EP, Los Gatos Research CO 23r, and ThermoScientific 42i-TL gas analyzers was provided by the Research Centre GEOMODEL of St. Petersburg State University (http://geomodel.spbu.ru/).

**Funding**

This activity has received funding from the European Union's Horizon 2020 research and innovation programme under grant
agreement No 776810 (VERIFY project). This work was supported by funding from the Helmholtz Association in the framework of MOSES (Modular Observation Solutions for Earth Systems). The development of the COCCON data processing tools were supported by ESA in the framework of the projects COCCON-PROCEEDS and COCCON-PROCEEDS II. The research was supported by Russian Foundation for Basic Research through the project No. 18-05-00011.

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

**Table 1. Area fluxes for $CO_2$ (kt km$^{-2}$ yr$^{-1}$), $CH_4$ (t km$^{-2}$ yr$^{-1}$), $CO$ (t km$^{-2}$ yr$^{-1}$) and $NO_x$ (t km$^{-2}$ yr$^{-1}$) obtained during EMME-2019 and the flux estimates for St. Petersburg based on in situ measurements. The values previously reported in literature are also presented.**

| Area flux | EMME | | In situ measurements | Literature sources | |
|---|---|---|---|---|---|
| | (9 days) | (4 days) | | St. Petersburg | The world's cities |
| **1** | **2** | **3** | **4** | **5** | **6** |
| $CO_2$, kt km$^{-2}$ yr$^{-1}$ | 89 ± 28 | 85 ± 12 | 40 ± 30 | 31 (Serebritsky, 2018), 46 (EDGAR database, 2018) 6 (suburbs, Makarova, 2018) | 29 (London, O'Shea, 2014) 35.5 (London, Helfter, 2011) 12.8 (Mexico City, Velasco, 2005) 12.3 (Tokyo, Moriwaki and Kanda, 2004) 0.8 – 7.7 (Krakow, Zimnoch, 2010) 28.3 (Berlin, Hase, 2015) |
| $CH_4$, t km$^{-2}$ yr$^{-1}$ | 135 ± 68 | 178 ± 30 | 120 ± 80 | 25 (Serebritsky, 2018, 2019), 110 (Makarova, 2006), 44 (suburbs, Makarova, 2018) 32 (suburbs, Zinchenko, 2002) | 66 (London, O'Shea, 2014) 7 – 28 (Krakow, Zimnoch, 2010) |
| $CO$, t km$^{-2}$ yr$^{-1}$ | 251 ± 104 | 333 ± 103 | 90 ± 50 | 410 (Serebritsky, 2018, 2019), 390 (Makarova, 2011), 90 (suburbs, Makarova, 2018) | 106 (London, O'Shea, 2014) 1520 (Mexico City, Stremme, 2013) |
| $NO_x$, t km$^{-2}$ yr$^{-1}$ | 66 ± 28 | - | - | 69 (Serebritsky, 2018, 2019) | 63-252 (London, Lee, 2015) 13- 300 (Norfolk, Marr, 2013) |


**Table 2. Emission ratios _ER_, obtained during EMME-2019 and the _ER_ estimates for St. Petersburg based on in situ measurements. The values previously reported in literature are also presented. In columns 2, 3, and 4 the values of the correlation coefficient (_r_) for corresponding datasets are given in parentheses.**

| Emission ratio | St. Petersburg | | | | Literature sources |
|---|---|---|---|---|---|
| | EMME | | In situ measurements | Official emission inventory | |
| | (9 days) | (4 days) | | | |
| 1 | 2 | 3 | 4 | 5 | 6 |
| $CO/CO_2$, ppbv/ppmv | 5.9 (r=0.88±0.02) | 6.2 (r=0.97±0.01) | 6.0 ± 2.4 (r=0.76±0.04) | 21 (Serebritsky, 2018, 2019) | 5.68, 8.44 (Paris, Ammoura, 2014), 1.92 – 6.6 (London, O'Shea, 2014), 6-9 (Indianapolis, Turnbull, 2015) 14 (Sacramento, Turnbull, 2011) |
| $CH_4/CO_2$, ppbv/ppmv | 6.8 (r=0.82±0.03) | 5.8 (r=0.96±0.02) | 7.8 ± 2.6 (r=0.70±0.04) | 2.2 (Serebritsky, 2018, 2019) | 3.9 - 6.9 (London, O'Shea, 2014), 5.2 ± 0.5 (London, Helfter, 2011), |

**Table 3. Comparison of the gas concentrations in air samples collected at the surface and elevated levels on 24 April 2019 and 25 April 2019 at the locations of FTS measurements inside the city plume.**


| Gas | 24 April 2019 (location B2) | | 25 April 2019 (location A5) | |
|---|---|---|---|---|
| | Surface level | Kite (~100 m) | Surface level | Kite (~70 m) |
| NO [mkg m-3] | 0 | 0 | 6 | 5 |
| $NO_2$ [mkg m-3] | 26.5 | 23.5 | 138.1 | 122.4 |
| $CH_4$ [ppmv] | 1.958 | 1.959 | 2.338 | 2.278 |
| $CO_2$ [ppmv] | 422.5 | 417.1 | 444.0 | 445.0 |
| CO [ppbv] | 191.1 | 185.8 | - | - |

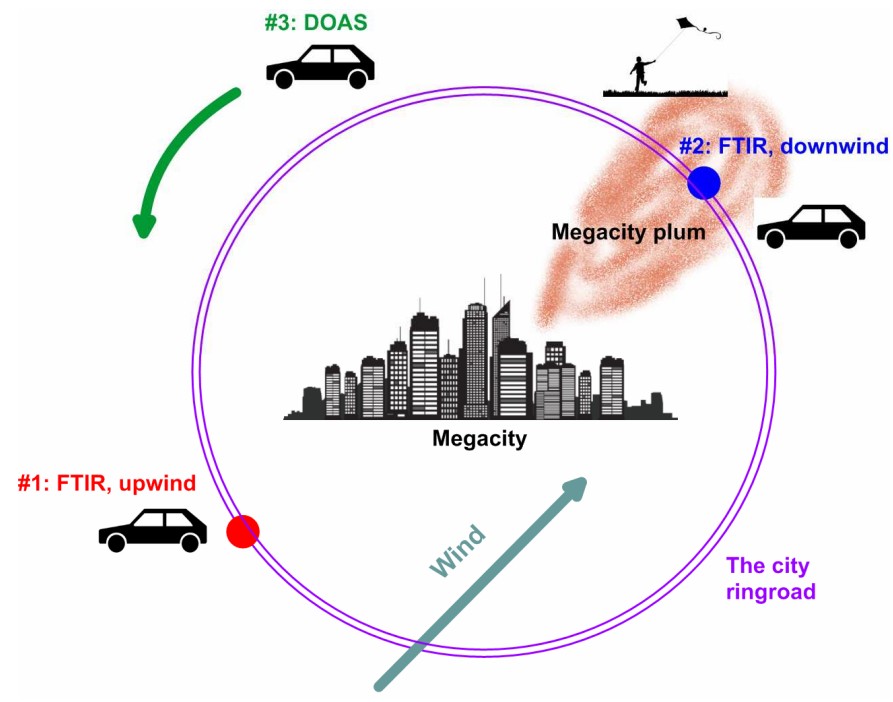


**Figure 1: Illustration of the concept of EMME: two FTIR spectrometers at the upwind and downwind locations on the opposite sides of the city (#1 and #2, red and blue dots) and circular moving DOAS technique spectrometer (#3). Ground-level air samples were collected at locations #2 and #3. Collecting air portions with the help of a kite was done usually at the downwind location under suitable weather and landscape conditions. Pictogram png-images: https://www.cleanpng.com/, last access 6 November 2019.**



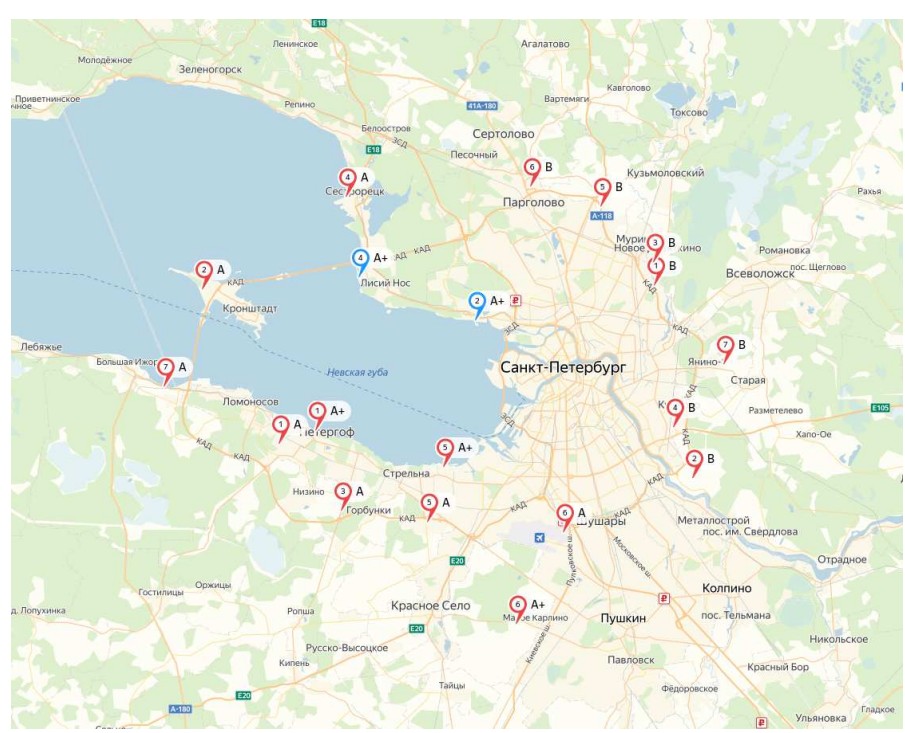

**Figure 2: The set of FTS locations around the St. Petersburg agglomeration. Locations are marked by letters "A" and "B" with numbers. The "plus" sign near a location mark denotes that there is a possibility to use local power supply at this location. Red colour denotes primary locations, blue colour denotes secondary locations. Map data © 2019 Yandex.**



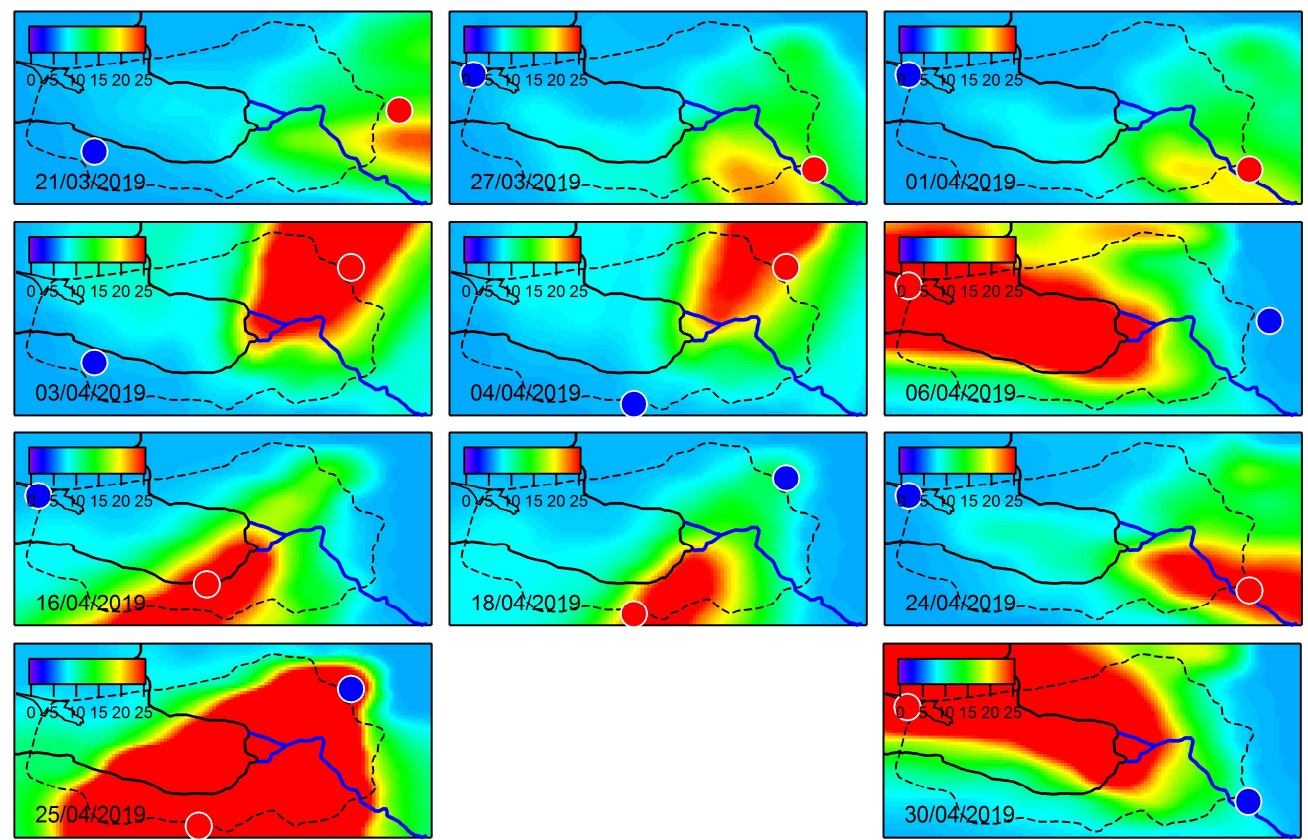

**Figure 3: The HYSPLIT model output for each of the campaign days (10:00 UTC) used as the forecast of the megacity plume while planning the field campaign. The colour bar units for TC$_{NO2}$ are [0-25] 10$^{15}$ cm$^{-2}$. The blue line in the southeast indicates the river Neva.**


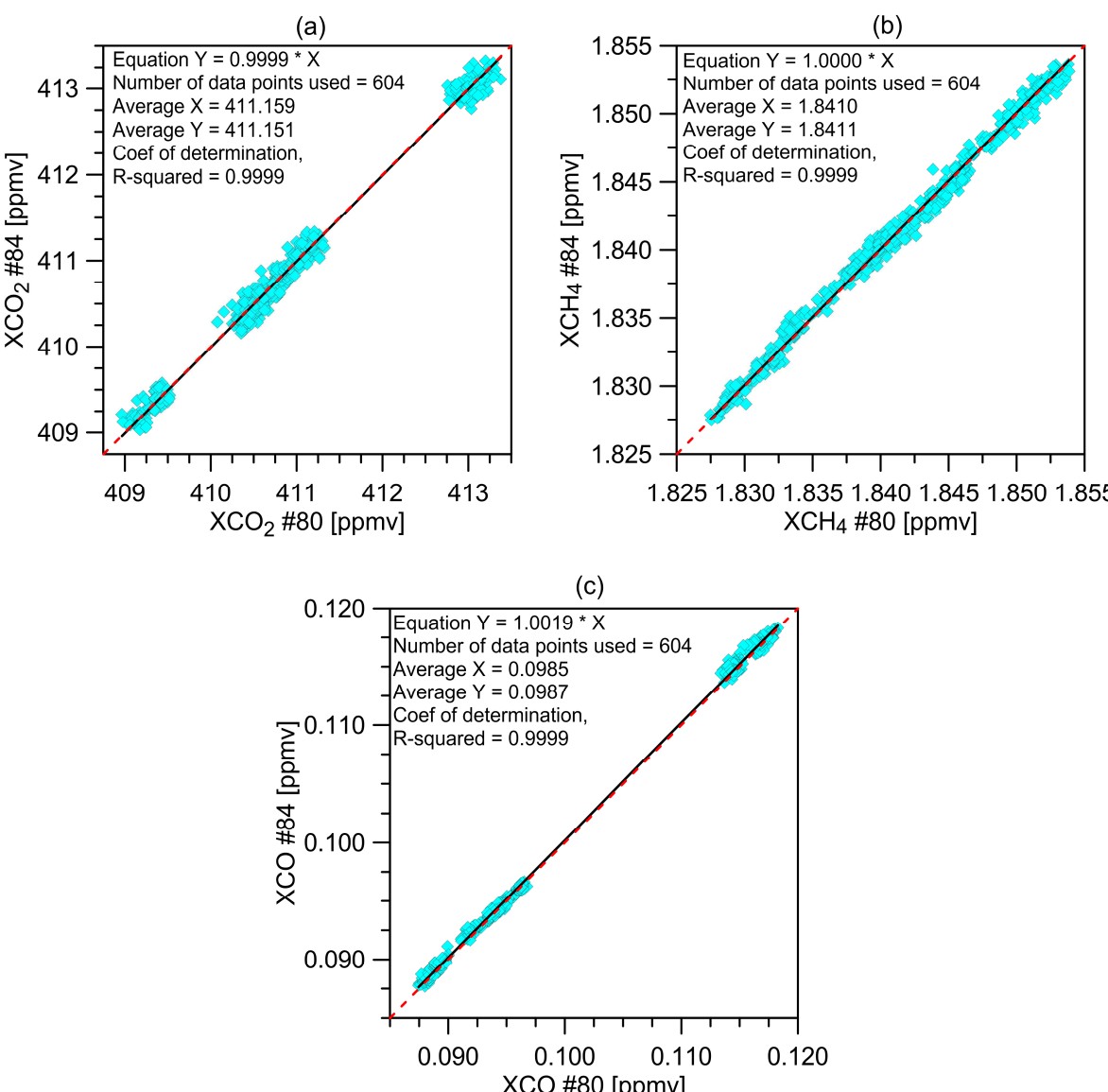

**Figure 4: The scatter plots of cross-comparison of the average mole fraction data during side-by-side calibrations.**


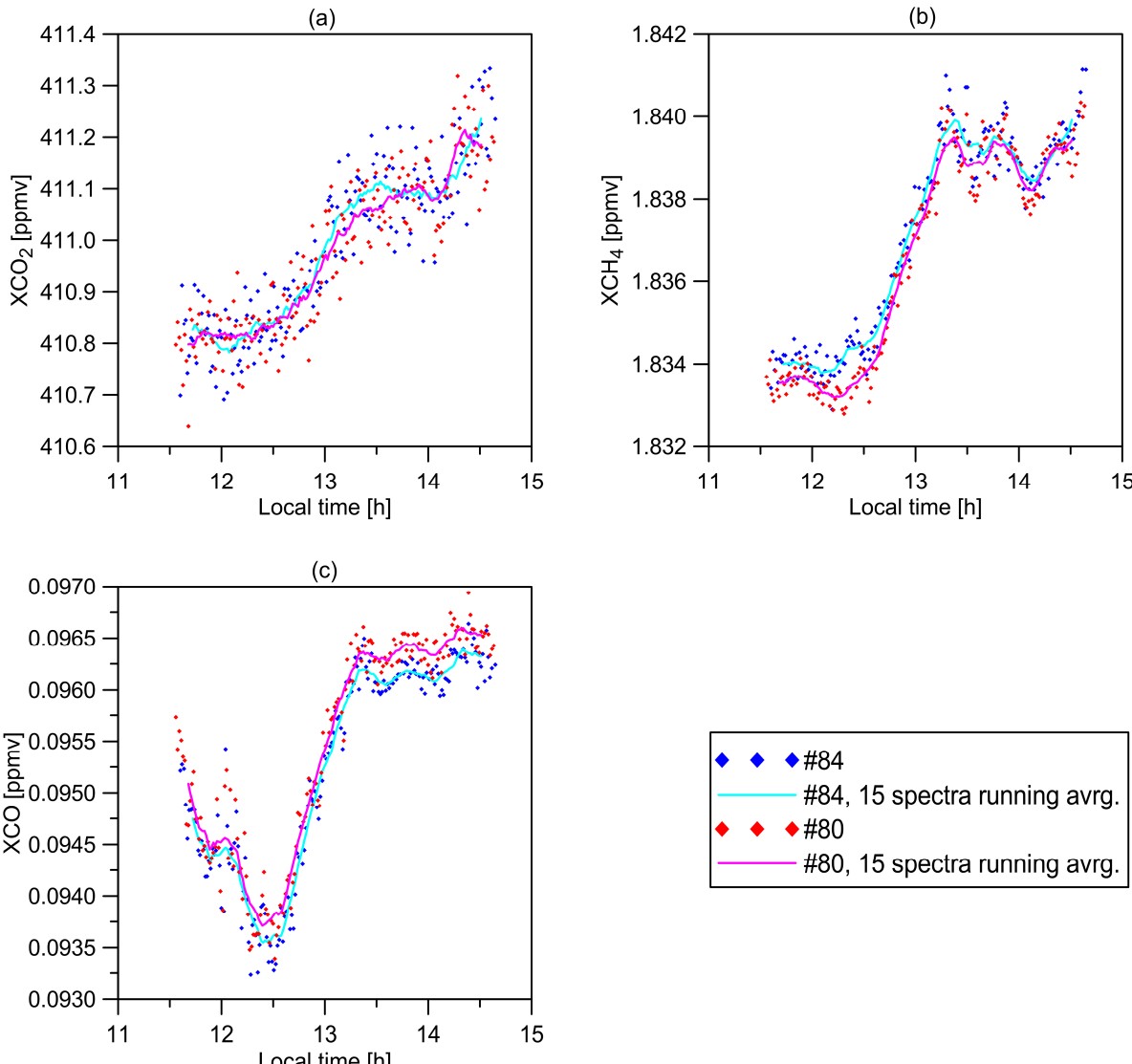


**Figure 5: The scaled results of the side-by-side measurements of XCO₂, XCH₄, and XCO by FTS#80 and FTS#84 on 12 April 2019.**

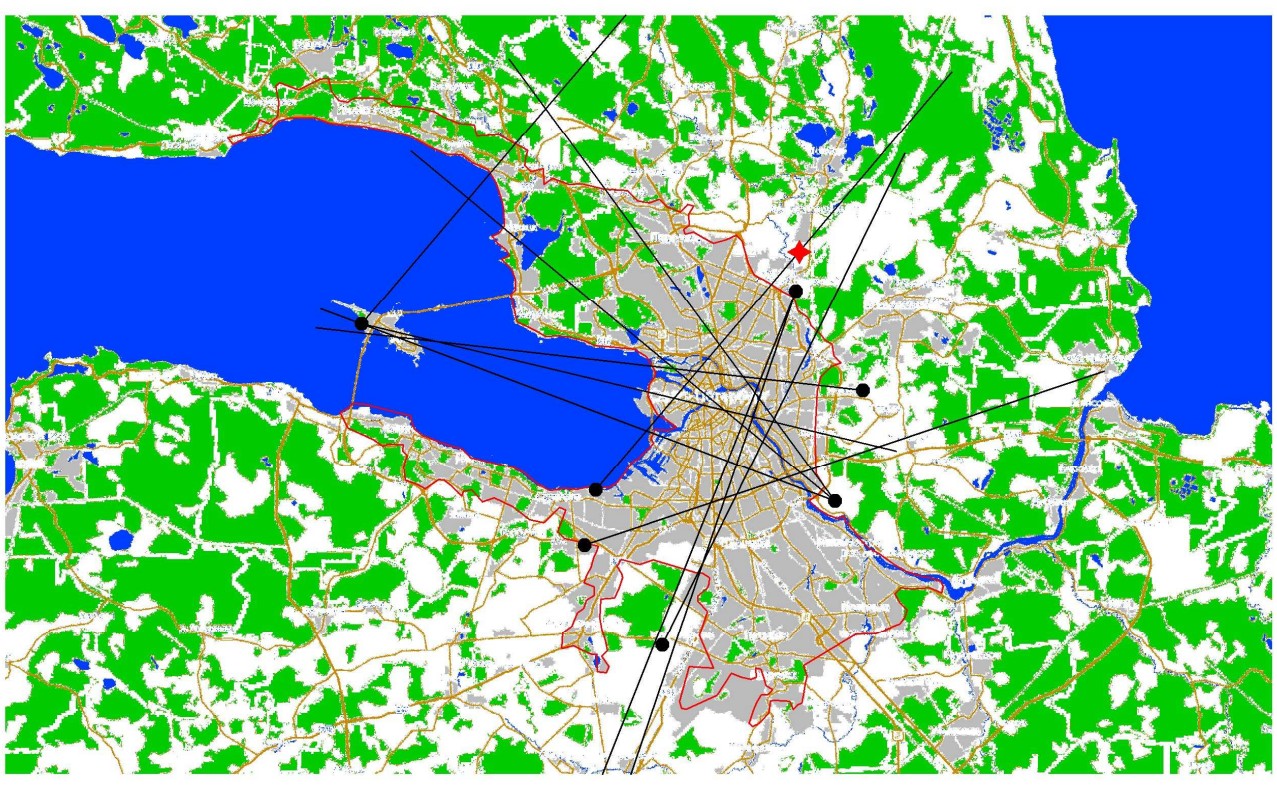


**Figure 6: An example of linear backward paths (black straight lines, black dots show the downwind FTS locations) for the days of FTIR observations. The major land use classes are shown by different colours (blue for the water bodies, grey for the residential buildings/industrial areas, green for the parks and forests). The path lengths on the map are plotted equal only for illustrative purpose. In fact they are all different since the FTIR observation locations and the wind field change from day to day. Red line**
**designates the official administrative boundary of the St. Petersburg agglomeration. Red "star" indicates the location of one of the major thermal power stations (TPS) located to the north of St. Petersburg. Map data © 2019 Yandex.**


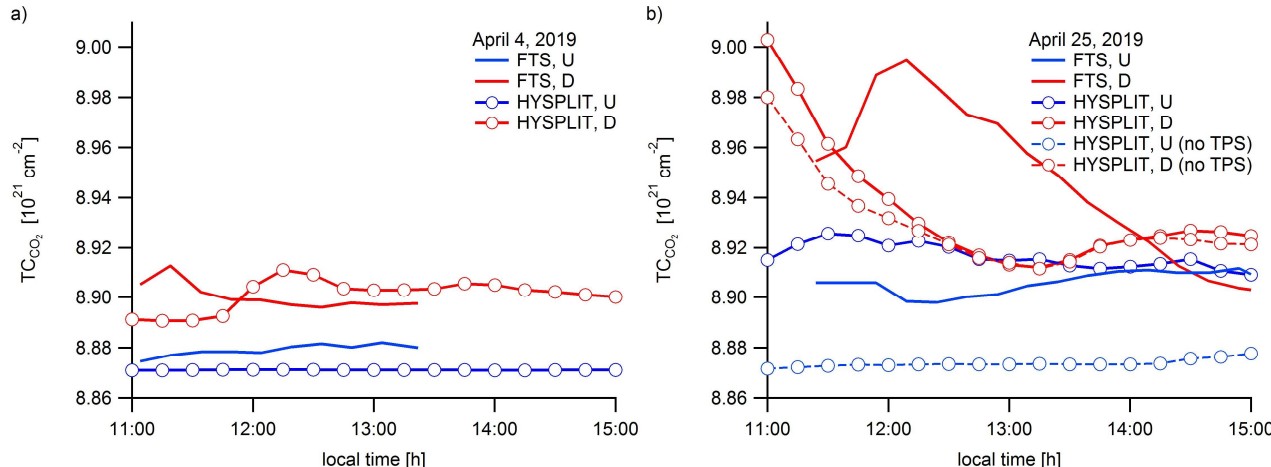

Figure 7: Time series of the $CO_2$ TC measurements by mobile FTS at upwind (U, blue) and downwind (D, red) locations on two days, April 4 and April 25, 2019. The measurements are compared with the results of the HYSPLIT simulations at both locations, upwind and downwind. For the day of April 25, special HYSPLIT scenario is added for comparison: the emission of the major thermal power station (TPS) of St. Petersburg nearby the upwind FTS location is turned off ("no TPS", see Fig. 8 and the text for details).


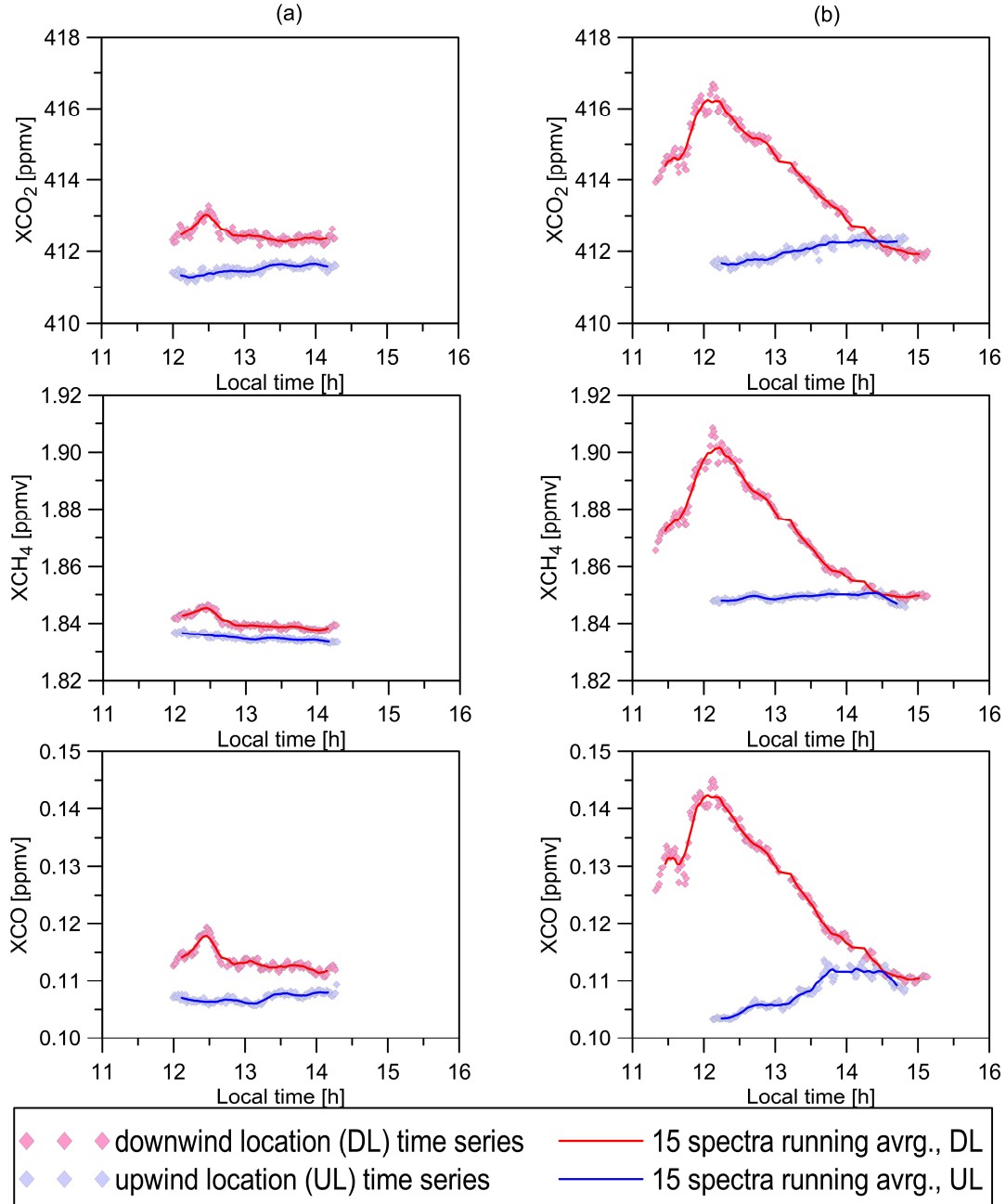

**Figure 8: Time series of Xgas for 4 April (a) and 25 April (b) at the clean location of FTS (blue dots) and at the polluted location of FTS (red dots). Solid lines of corresponding colours denote 15 min running average.**

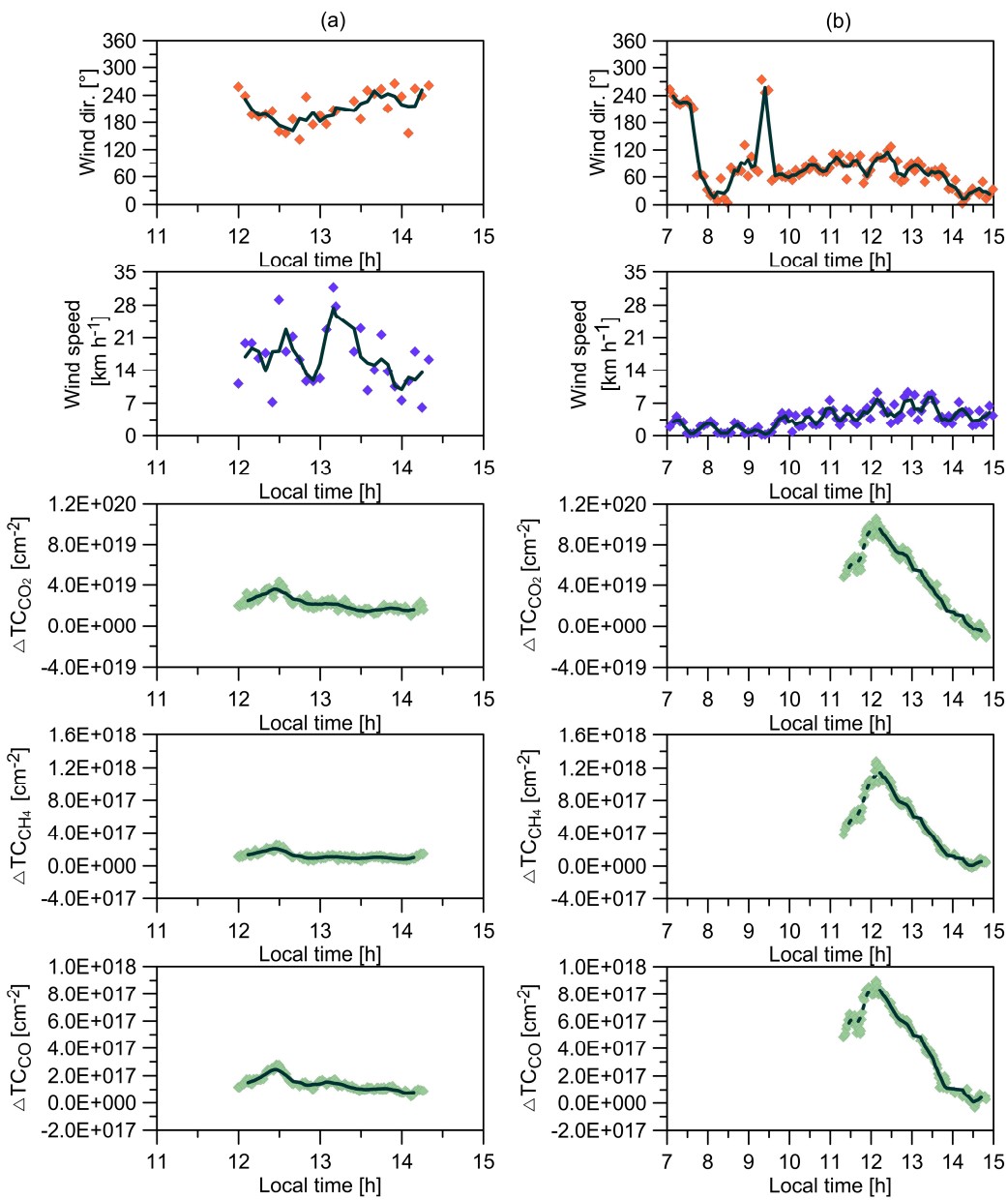


**Figure 9: The difference between the TC values at the polluted and clean locations of FTS on 4 April (a) and 25 April (b). The wind speed and direction are also shown.. Solid lines denote 15 min running average. Dashed lines denote time interval when extrapolated input data from the clean location were used (see text).**

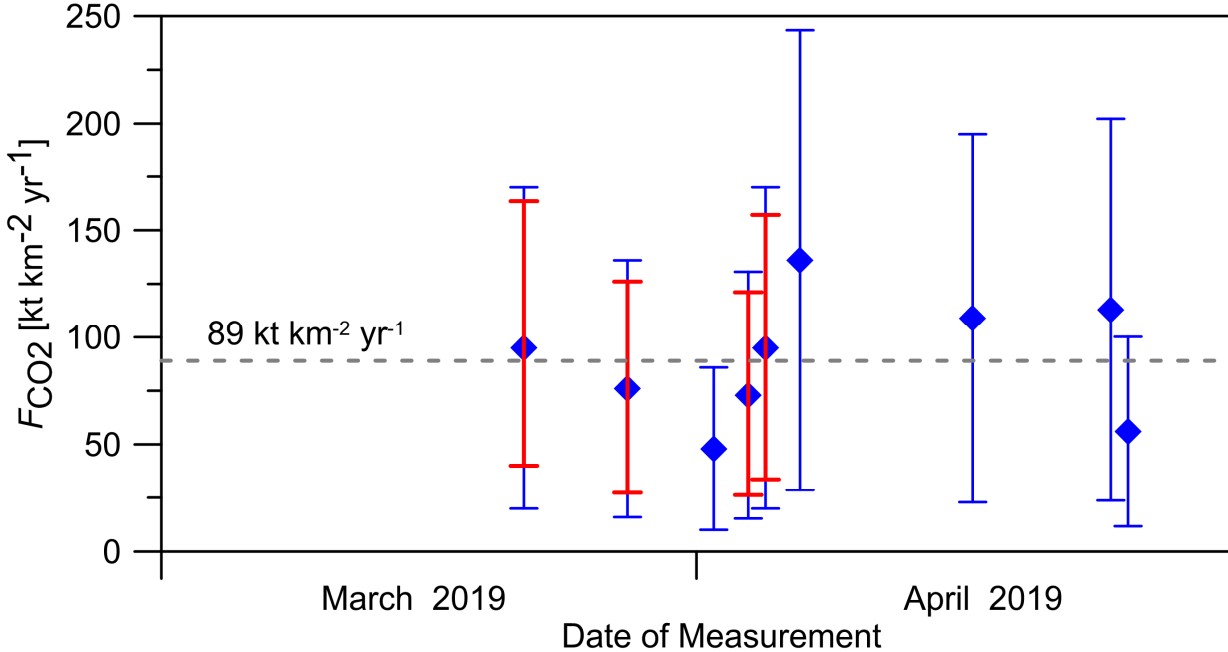

 **Figure 10: Daily mean values of the CO₂ area flux *F* obtained during the city campaign. Error bars show the uncertainties of *F* values estimated for the 9-day and 4-day data sets (blue and red respectively).**

## Appendix A: Description of the experiment details and meteorological conditions

Table A1 contains information for all days of the field campaign such as the location of FTIR spectrometers, FTIR spectrometer identifier, number of bags of air samples, flight of a kite and air sampling altitude. The last column of Table A1 includes information on the experiment setup (up-and downwind or cross sectional setup) and FTIR spectrometer operator's notes about meteorological phenomena, changes in cloud cover, and local air pollution events observed during FTIR field measurements.

In Table A2 we collect the main characteristics of weather conditions for each measurement day. The weather information is provided for local noon from the observational data of the meteorological station located in the centre of St. Petersburg (index no. 26063, 59.97°N, 30.28°E). The daytime surface air temperature was varying from ~0 °C on March 27 to +21 °C on April 25; relative humidity – varying from 84% on March 21 to 21% in April 6. Generally, surface wind speed throughout the campaign was moderate in the range of 2-3 m s-1, except on April 24 and 25, when light surface winds were registered (1 m s $^{-1}$). Prevailing wind direction for St. Petersburg is southwest, and surface winds blowing from southwest and west-southwest were registered during most days of the campaign; however, other wind directions were registered, too (see Table A2). An average wind is calculated for the time period of FTIR observations. Resulting wind speeds and directions from the three different data sources are given in Table A3.

The satellite images of cloud cover detected by the MODIS satellite instrument in the vicinity of St. Petersburg are presented in Fig. A1. They confirm daytime clear sky conditions for the duration of the campaign, except the day of April 30, when the altocumulus translucidus clouds started to develop.

**Table A1. EMME-2019 observation details: the field experiment setup (up- and downwind "u&d" or cross sectional "cs"), the FTS location (Loc), the FTS identifier (FTS#), the number of bags of air samples (AS), indication of the kite launch and the corresponding air sampling altitude.**

| Date of 2019 | Outside the city plume | | | | Inside the city plume | | | | DOAS mobile | Comment |
|---|---|---|---|---|---|---|---|---|---|---|
| | Loc | FTS# | AS | Kite | Loc | FTS# | AS | Kite | | |
| 21.03 | A1 | #80 | 2 | no | B7 | #84 | 2 | yes | no | U&d setup,test FTIR field measurements, test flight of the kite without air sampling |
| 27.03 | A2 | #84 | 2 | no | B2 | #80 | 2 | no | yes | U&d setup, A2 – no clouds, B2 – groups of clouds |
| 01.04 | A2 | #84 | 2 | no | B2 | #80 | 2 | no | yes | U&d setup, A2 – no clouds, B2 – groups of clouds |
| 03.04 | A1 | #84 | 2 | no | B3 | #80 | 2 | no | yes | U&d setup, clear sky for both locations |
| 04.04 | A5 | #84 | 2 | no | B3 | #80 | 2 | no | yes | U&d setup, clear sky for both locations |
| 06.04 | B7 | #84 | 2 | no | A2 | #80 | 2 | no | no | U&d setup, clear sky and burning grass for both locations |
| 16.04 | A2 | #84 | 2 | no | A5+ | #80 | 2 | no | yes | Cs setup, clear sky for both locations |
| 18.04 | B3 | #80 | 2 | no | A5, A6+ | #84 | 2 | no | yes | U&d setup, clear sky for both locations |
| 24.04 | A2 | #84 | 2 | no | B2 | #80 | 2 | Yes, 100 m | yes | U&d setup, A2 – clear sky, B2 – light cirrostratus, sun halo |
| 25.04 | B3 | #80 | 2 | no | A5 | #84 | 2 | Yes, 70 m | yes | U&d setup, B3 – smoke plum in the field of view of FTIR spectrometer, A5 – light cirrostratus |
| 30.04 | B2 | #80 | 2 | no | A2 | #84 | 2 | no | yes | U&d setup, B2 – cirrostratus, A2 – quickly developing altocumulus translucidus |

**Table A2. Basic meteorological data for the days of the field campaign: surface air temperature (T), relative humidity (RH), wind speed (WS) and wind direction (WD) at local noon. The meteorological data refers to one of the observational sites in the city of St. Petersburg (http://rp5.ru/Weather_archive_in_Saint_Petersburg, last access 5 March 2020).**

| Date | T (ºC) | RH (%) | WD | WS (m s$^{-1}$) |
|---|---|---|---|---|
| 21 March (Th) | 2.3 | 84 | WSW | 3 |
| 27 March (We) | 0.1 | 64 | WSW | 2 |
| 1 April (Mo) | 3.2 | 76 | WSW | 3 |
| 3 April (We) | 9.8 | 24 | S | 3 |
| 4 April (Th) | 12.5 | 24 | SW | 3 |
| 6 April (Sa) | 12.5 | 21 | SE | 2 |
| 16 April (Su) | 12.0 | 39 | NE | 2 |
| 18 April (Tu) | 12.5 | 35 | NE | 2 |
| 24 April (We) | 16.7 | 40 | WSW | 1 |
| 25 April (Th) | 20.9 | 23 | WSW | 1 |
| 30 April (Tu) | 10.7 | 27 | SSE | 2 |


**Table A3. The wind speed and the wind direction for the days of the field campaign, as retrieved from different data sources: in situ observations (LOCAL), globally gridded assimilated data (GDAS) and backward trajectory calculations (HYSPLIT).**


| Date | Wind speed, m s$^{-1}$ | | | Wind direction, ° | | |
|---|---|---|---|---|---|---|
| | LOCAL | GDAS | HYSPLIT | LOCAL | GDAS | HYSPLIT |
| 21 March | 6 | 7 | 10 | 293 | 270 | 277 |
| 27 March | 2 | 5 | 5 | 292 | 332 | 324 |
| 1 April | 3 | 5 | 8 | 329 | 307 | 310 |
| 3 April | 3 | 5 | 5 | 212 | 193 | 199 |
| 4 April | 3 | 6 | 6 | 214 | 194 | 202 |
| 6 April | 1 | 3 | 3 | 58 | 104 | 103 |
| 16 April | 1 | 5 | 6 | 36 | 42 | 40 |
| 18 April | 1 | 5 | 7 | 25 | 34 | 26 |
| 24 April | 3 | 5 | 6 | 357 | 286 | 291 |
| 25 April | 1 | 2 | 1 | 69 | 95 | 71 |
| 30 April | 2 | 4 | 4 | 78 | 112 | 40 |

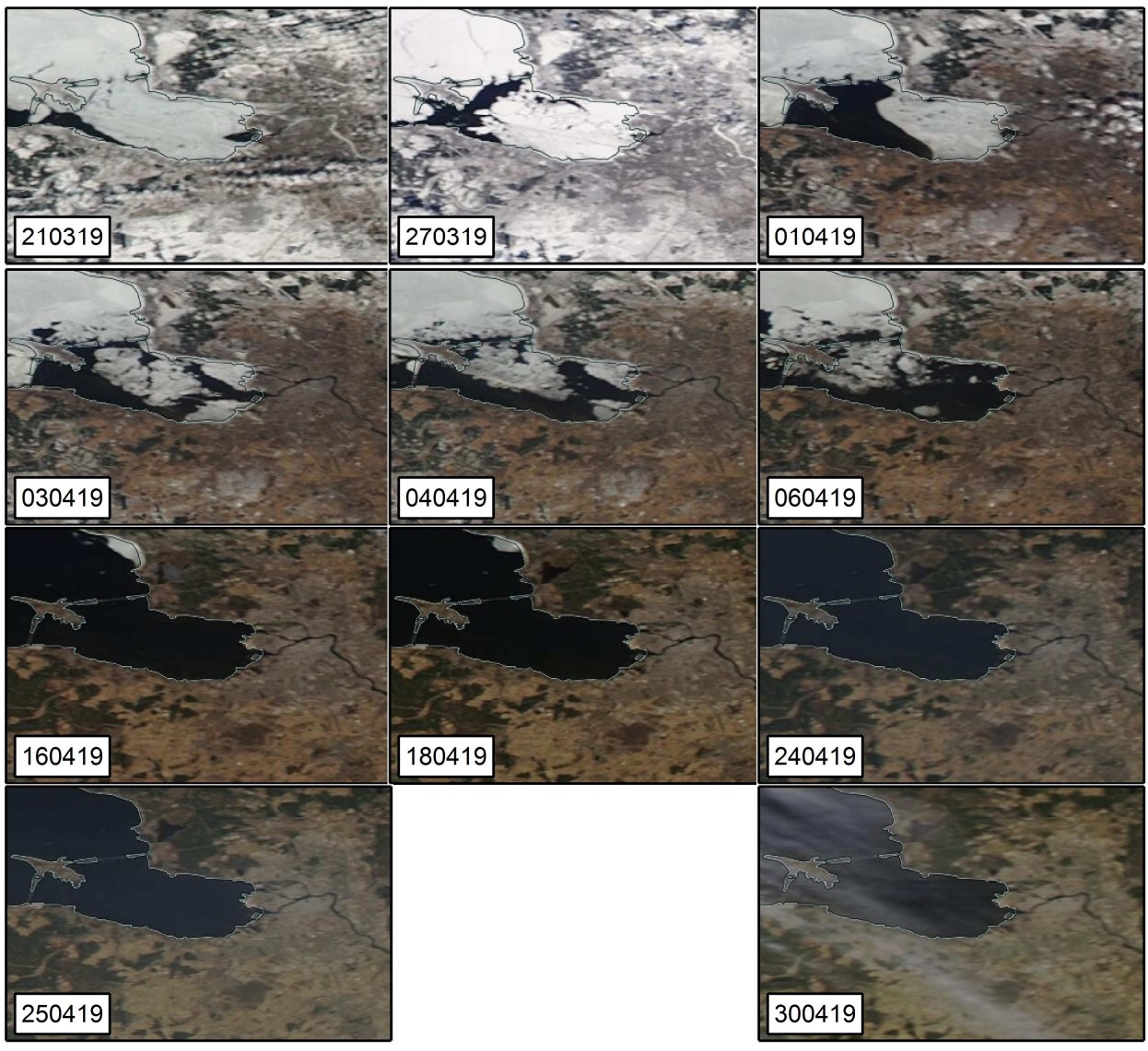


**Figure A1: The MODIS satellite images of cloud cover in the vicinity of St. Petersburg taken on the days of field campaign.**

## Appendix B: Area fluxes for simplified box model setup

Area fluxes for $CO_2$, $CH_4$, CO and $NO_x$ estimated using the simplified box model setup with a constant path length ($L_j(t_i)=L=$const$\approx$27 km for each day of field observations) are given in Table B1.

**Table B1. Area fluxes for $CO_2$ (kt km$^{-2}$ yr$^{-1}$), $CH_4$ (t km$^{-2}$ yr$^{-1}$), CO (t km$^{-2}$ yr$^{-1}$) and $NO_x$ (t km$^{-2}$ yr$^{-1}$) obtained using**

 **constant path length approach.**

| Area flux | EMME | | In situ measurements |
|---|---|---|---|
| | (9 days) | (4 days) | |
| **1** | **2** | **3** | **4** |
| $CO_2$, kt km$^{-2}$ yr$^{-1}$ | 96 ± 25 | 99 ± 17 | 32 ± 27 |
| $CH_4$, t km$^{-2}$ yr$^{-1}$ | 151 ± 82 | 213 ± 57 | 95 ± 64 |
| CO, t km$^{-2}$ yr$^{-1}$ | 276 ± 117 | 385 ± 97 | 71 ± 40 |
| $NO_x$, t km$^{-2}$ yr$^{-1}$ | 74 ± 30 | - | - |

**Appendix C: Comments on transport of the pollutants from elevated sources**

We illustrate transport of the pollutants from elevated sources with a HYSPLIT simulation (see Fig. C1). We selected one of the days of EMME (April 16, 2019) and simulated the $CO_2$ emission from a 180-meter chimney of the thermal power station mentioned above in the main text of the article. The plot presents a 34-hour trajectory of the mass-weighted $CO_2$ plume position (the centroid of the plume) on the geographical map (top panel) and using the altitude scale (bottom panel). One can see that the plume centroid starts its movement from the chimney location at ~180 m altitude (12:00 of April 15) and raises

up to ~500 m in one hour; then it does not fall below the level of ~350 m during its "flight" length of more than 300 km. The detailed analysis of respective vertical profiles of $CO_2$ concentration shows its maximum at ~500 m, being 1.2 times higher than that on the surface at start and 3.6 times higher than that on the surface at the end of the plume trajectory. Thus, the probability to register high concentrations corresponding to the centroid of the plume by surface-based observations can be estimated as very low. Moreover, polluted air mass from a chimney is more likely to rise up, rather than descend to the

ground due to two reasons: (1) the vertical velocity of the air pollution jet emitted from a chimney can be rather high; (2) the temperature of a plume released from the chimney is usually significantly higher than the temperature of the ambient air causing the buoyancy effect.

     Elevated air sampling using kite launches was performed only twice during the EMME campaign, therefore the results of these kind of measurements could not be considered as a reliable confirmation of the absence of elevated plumes. The

presence of the elevated plumes of CO and $CO_2$ could be also confirmed by the following evidence. The comparison of the values of area fluxes ($F$, see Table 1) estimated using in-situ measurements (column #4) and FTIR observations (column #2 and #3) shows that for $CH_4$ which sources are mainly located on the ground surface we obtain significantly lower difference in corresponding $F$ values than for CO an $CO_2$.

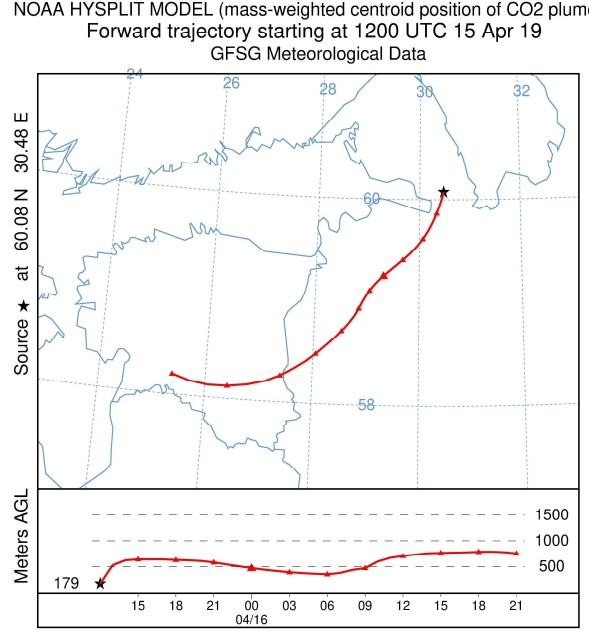


**Figure C1: Evolution of the mass-weighted centroid position of the CO2 plume taken as an example (see text).**