# Peer review of "Emission Monitoring Mobile Experiment (EMME): an overview and first results of the St. Petersburg megacity campaign-2019"

_Atmospheric Measurement Techniques, 2020_

## Referee Comment (RC1) · Anonymous Referee #1 · 8 Jul 2020

The paper "Emission Monitoring Mobile Experiment (EMME): an overview and first results of the St. Petersburg megacity campaign-2019" by Makarova et al. presented the study conducted in St. Petersburg for GHG and NOx emission assessments. Different methods including the differential column measurements using two solar tracking spectrometers (EM27SUN), mobile DOAS measurements and in-situ measurements are combined to achieve this goal. Deploying the mass balance method, their obtained emission flux numbers of CO2 and CH4 are much higher compared to the numbers of the emission inventories. The descriptions of the campaign are well elaborated and data analysis is clearly structured. However, several comments needed to be addressed before the consideration of the publication.

1) The abstract presents a lot of technical details, such as the data processing activities in four steps. I recommend to remove these.

2) Part of the methodology is based on emission assessments using differential column measurements equipped with two solar-tracking spectrometers upwind and downwind of the city. The authors could consider to include Chen et al. (2016): "Differential column measurements using compact solar-tracking spectrometers", where the same principle has been used, as a reference in line 100.

3) Page 9: The authors have determined the optimum integration time by examining the "half width" of the short term variations. Another possibility to determine the optimum integration time is to use the Allan variance analysis. This approach was used in Chen et al. (2016).

4) Page 9: please add units to the parameters denoted in equation (1).

5) Section 4.4: I have doubts about the definition of the effective air parcel path length. By deriving the effective path length including only the "polluted path", and excluding the "clean path", you are determining the emission flux of the industrial and traffic (the polluted areas), but not the emission flux of the whole city. So it could be not fair to compare these numbers to the emission inventories of the city, which may result in much higher emissions compared to the emission inventory.

6) Line 358: repetition of "April 25", please delete the second one.

7) Equation 2: It is not clear what kind of wind speeds are taken for the consideration, please elaborate it.

8) Equation 2: you can determine the square root of the error terms instead of adding them

9) Figure 5: there is no unit for the color bar [0-25]. The river is drawn as blue, but it looks confusing because the blue color is also assigned to the color bar.

10) Figure 7: you could show the scaled results instead. It will illustrate how the close the curves are to each other after the scaling process.

11) Figure 8: It is not very clear from the description which paths you took for determining the effective path length, are these paths from different days? Please elaborate these further. Do you have only one effective path length for all the days for each meteorological data set (LOCAL, GDAS, and HYSPLIT)? If so, how the effective path lengths vary given by different meteorological data set?

12) Table 4: The big discrepancies between the estimate in the paper and the emission inventory could be partially attributed to the usage of the effective path length, so the flux density determined in this study is focused on the industry area and traffics whereas the inventory is the averaged flux in the city. Please discuss this possibility.

---

## Referee Comment (RC2) · Anonymous Referee #2 · 24 Jul 2020

This paper describes a new top down approach of estimating the total emissions of several climate gases and air pollutants from a megacity. A similar approach has only been applied a few times in other cities and here the emissions for the full city of Sankt Petersburg are presented.

General comments

The paper is well written, with good language and nice, instructive graphs in most cases. It is claimed that the objective of the paper is to provide emission numbers for Sankt Petersburg. However a significant, and in my mind, to big part of the paper describes the general methodology with complementary data. The abstract is rather

long and detailed, and it should be made more concise with focus on the results. The main body is too detailed for a scientific paper: a) The Modis data is not relevant since it is not actively used, b) Remove nice photos of StPetersburg, c) In the introduction, there is a lot of explanation about different variants of obtaining windspeed and effective path, but this is not used in any significant extent in the results; this should dbe shortened

If I understand right, the methodology is the same as used in other campaigns (Berlin). In the introduction or elsewhere an overview about the other studies should be added with discussion on how comparable this study is to the other ones in terms of methodology and results . E.g. was effective path used by other studies. In Eq 1 you calculate the flux using total column (needed to get the right unit. You also introduce Xgas (I assume against total pressure). When do you use Xgas in the calculation? Is it only to show thing quantitatively? I assume in most cases te pressure is the same for up and downwind site ? Add in the text a definition of Xgas (not know for everyone) and describe what is your purpose here for showing it ?

For the wind used in the final results the authors rely on the Hysplit model, which in turn is based on a global model (NCEP) for the wind. The authors argue that the use of data from this model provides less variability in the final results. I argue that the wind variability is less for the Hysplit data than for real measurements, since it is large domain model, and Hysplit will therefore artificially smooth the wind data. This should be beter discussed by the authors. The authors present their flux estimation based on modelled effective path. Such an excercise provides useful data but it is hard for the reader to understand how the data was produced and its errors, since the data represents a combination of measurements and model. I suggest presenting also the purely measured data based on a constant path. For the effective path the authors claim they made a land use analysis and they refer to a public web site but there little information given in the paper and it is hard for the reader to understand the assumptions made here. Forinstance, I am missing an explanation about what are the

hypothesis about the detailed emission source categories and differentiation between species ($CO_2$, $CH_4$, $NO_2$). The species above orginate from different emission source categories; e.g $CH_4$ could partly come from the waterways (sewers and water canals) and pipelines rather than mobile and fixed combustion sources which are relevant for $CO_2$ and $NO_2$. This will make the effective path species dependent. The emissions from water ways could also be impacted by windspeed. I suggest adding a graph for the landuse model and include the model as complementary material for this paper. The $NO_2$ DOAS data are explained very briefly wrt to methodology and results. Did you use the same methodology as for the other species, even though you measure in a full circle around town. I suggesting describing the methodology in a better way and results. Did you use the $NO_2$ data to correct the FTIR measured data, if so clarify.

The treatment of uncertainties is all based on the obtained/measured variability of the parameters used to calculate the flux (total column, effective path and wind). In my mind this is an assessment of the random uncertainty. However there is no mentioning of systematic errors of any of these parameters. Please add a discussion about this and change absolute uncertainties to random uncertainty. In the $CO_2$ and CO data there is a factor of two difference between the column measured data and the one measured by in situ data. This is explained by the fact that the $CO_2$ and CO emissions are released from high chimneys (200m). However the mixing layer should be several hundred meters (at minimum) at solar conditions and the pollutants should therefore well mixed at some distances from the chimney (>1 km). This was also supported by kite measurements. In addition a considerable portion of the $CO_2$ should come from transport sector. The discussion should be improved on this topic. , All in all, I believe the paper should be published, with some minor improvements, based on my general and specific comments.

Specific comments

(Note that some of the comments below will be the same as in the general comments)

[Figure]

P3: Row 83: When making refence to other studies it would be relevant to add similar large scale measurements by mobile FTIR (Solar Occultation Flux technique) and mobile DOAS which has been applied fo large scale flux measurements for at least decade by now : e.g. 1. de Foy, et al., (2007) Modelling constraints on the emission inventory and on vertical dispersion for CO and SO2 in the Mexico City Metropolitan Area using Solar FTIR and zenith sky UV spectroscopy. Atmospheric Chemistry And Physics 7, pp. 781-801. DOI: 10.5194/acp-7-781-2007. 2. Mellqvist, et al., (2010) Measurements of industrial emissions of alkenes in Texas using the solar occultation flux method. Journal of Geophysical Research - Atmospheres 115. DOI: 10.1029/2008JD011682. 3. Johansson, J., et al. (2014) Emission measurements of alkenes, alkanes, SO2, and NO2 from stationary sources in Southeast Texas over a 5 year period using SOF and mobile DOAS. Journal of Geophysical Research-Atmospheres 119, no. 4, pp. 1973-1991. DOI: 10.1002/2013jd020485. 4. Johansson, et al. (2014) Quantitative measurements and modeling of industrial formaldehyde emissions in the Greater Houston area during campaigns in 2009 and 2011. Journal of Geophysical Research-Atmospheres 119, no. 7, pp. 4303-4322. DOI: 10.1002/2013JD020159. 5. Kille N, et al, The CU Mobile Solar Occultation Fluxinstrument, AMT, 10, 373-392, 2017

P 5, row 121: You claim that the DOAS measures tropospheric columns. Please elaborate in a few sentences what is actually measured, even though you refer to previous studies. Are you using multiaxis measurements to derive absolute columns or is it differential columns assuming that the upwind measurements is free from troposheric NO2, and hence that the differential measurements corresponds to the tropospheric absolute column

P5, row 132. Add references from other places on mobile DOAS, e.g. Johansson, M et al., Mobile mini-DOAS measurement of the outflow of NO2 and HCHO from Mexico city, ACP, 9(15):5647-5653, 2009. Rivera, C. et al., (2010) Quantification of NO2 and SO2 emissions from the Houston Ship Channel and Texas City industrial areas during the 2006 Texas Air Quality Study. Journal of Geophysical Research - Atmospheres

115. DOI: 10.1029/2009JD012675.

P6, row 171: This sentence is unclear rewrite it. Forinstance Table 1 presents daily information ...

P8, row 128: Define Xgas (is it against pressure?) and motivate why you introduce this. Would it not be more appropiate to compare total columns instead of Xgas since TC is the ones used for the flux.

P8, row 232: The comparions between the two spectrometers is very convincing. Nevertheless, it only shows how the spectral properties of two spectrometers influences the statistical error of the measurements. Please comment how this information was used.

P 9, 244: I think this section should be more detailed wrt the spectroscopy. At least a couple of general sentences for how te retrieval is done and if there are interfering species etc could be helpful,

---

## Author Comment (AC1) · 29 Sep 2020

**The reply to the anonymous referee #1 (RC1)**

We are grateful to the referee for the very insightful comments. We took them into account while preparing the revised version of the manuscript.

Below, the actual comments of the referee are given in **bold courier font and blue colour**.
The text added to the revised version of the manuscript is marked by red colour.

**1) The abstract presents a lot of technical details, such as the data processing activities in four steps. I recommend to remove these.**
The text about four steps of data processing has been removed from the abstract.

**2) Part of the methodology is based on emission assessments using differential column measurements equipped with two solar-tracking spectrometers upwind and downwind of the city. The authors could consider to include Chen et al. (2016): "Differential column measurements using compact solar-tracking spectrometers", where the same principle has been used, as a reference in line 100.**

We added the suggested reference in several places, in particular:

> Chen et al. (2016) developed and used differential column methodology (downwind-minus-upwind column differences) for the evaluation of $CH_4$ emissions from dairy farms in the Chino area.
>
> ………….
> The idea and the methodology of EMME experiment were based mainly on the studies by Hase et al. (2015), Ionov and Poberovskii (2015), Chen et al. (2016) and Viatte et al. (2017).

**3) Page 9: The authors have determined the optimum integration time by examining the "half width" of the short term variations. Another possibility to determine the optimum integration time is to use the Allan variance analysis. This approach was used in Chen et al. (2016).**

We added the following text at the end of Section 4.2:

> The chosen averaging interval of 15 min is in good agreement with the estimation of the optimal integration time (10 min) obtained as a result of the Allan analysis implemented by Chen et al. (2016). Chen et al. (2016) applied this approach to the differential measurements of XCO2, XCH4 performed by three EM27/SUN spectrometers within urban areas.

**4) Page 9: please add units to the parameters denoted in equation (1).**

In the revised version units are added to the parameters denoted in equations (1-3).

**5) Section 4.4: I have doubts about the definition of the effective air parcel path length. By deriving the effective path length including only the "polluted path", and excluding the "clean path", you are determining the emission flux of the industrial and traffic (the polluted areas), but not the emission flux of the whole city. So it could be not fair to compare these numbers to the emission inventories of the city, which may result in much higher emissions compared to the emission inventory.**

**+**

**12) Table 4: The big discrepancies between the estimate in the paper and the emission inventory could be partially attributed to the usage of the effective path length, so the flux density determined in this study is focused on the industry area and traffics whereas the inventory is the averaged flux in the city. Please discuss this possibility.**

The main goal of the field campaign is to evaluate the area fluxes (F) originated from the urbanized territories of the St.Petersburg agglomeration. Therefore we excluded from the consideration the territories of parks, forests and water bodies as the areas that practically have no anthropogenic emission sources. At the same time we agree with the referee's statement that "**So it could be not fair to compare these numbers to the emission inventories of the city, which may result in much higher emissions compared to the emission inventory**". In the revised version of the manuscript, we estimated the urbanized area of the St.Petersburg agglomeration according to the land-use classification that was developed for the derivation of the effective path lengths. We obtained that the total urbanized area of the agglomeration occupies about 984 $km^2$ while the official area of the entire St.Petersburg is 1439 $km^2$. Therefore the values of area fluxes for all gases ($CO_2$, $CH_4$, CO and $NO_2$) that were estimated using the official inventory data have been recalculated and, as a result, became higher. Revised version of Table 1 (the former Table 4) is given below. The changes are highlighted by yellow colour.

**Table 1. Area fluxes for $CO_2$ (kt km$^{-2}$ yr$^{-1}$), $CH_4$ (t km$^{-2}$ yr$^{-1}$), CO (t km$^{-2}$ yr$^{-1}$) and $NO_x$ (t km$^{-2}$ yr$^{-1}$) obtained during EMME-2019 and the flux estimates for St. Petersburg based on in situ measurements. The values previously reported in literature are also presented.**

| Area flux | EMME | | In situ measurements | Literature sources | |
|---|---|---|---|---|---|
| | (9 days) | (4 days) | | St. Petersburg | The world's cities |
| **1** | **2** | **3** | **4** | **5** | **6** |
| $CO_2$, kt km$^{-2}$ yr$^{-1}$ | 89 ± 28 | 85 ± 12 | 40 ± 30 | 31 (Serebritsky, 2018), 46 (EDGAR database, 2018) 6 (suburbs, Makarova, 2018) | 29 (London, O'Shea, 2014) 35.5 (London, Helfter, 2011) 12.8 (Mexico City,Velasco, 2005) 12.3 (Tokyo, Moriwaki and Kanda, 2004) 0.8 – 7.7 (Krakow, Zimnoch, 2010) 28.3 (Berlin, Hase, 2015) |
| $CH_4$, t km$^{-2}$ yr$^{-1}$ | 135 ± 68 | 178 ± 30 | 120 ± 80 | 25 (Serebritsky, 2018, 2019), 110 (Makarova, 2006), 44 (suburbs, Makarova, 2018) 32 (suburbs, Zinchenko, 2002) | 66 (London, O'Shea, 2014) 7 – 28 (Krakow, Zimnoch, 2010) |
| CO, t km$^{-2}$ yr$^{-1}$ | 251± 104 | 333 ± 103 | 90 ± 50 | 410 (Serebritsky, 2018, 2019), 390 (Makarova, 2011), 90 (suburbs, Makarova, 2018) | 106 (London, O'Shea, 2014) 1520 (Mexico City, Stremme, 2013) |

| NO$_x$, t km$^{-2}$ yr$^{-1}$ | 66 ± 28 | - | - | 69 (Serebritsky, 2018, 2019) | 63-252 (London, Lee, 2015) 13- 300 (Norfolk, Marr, 2013) |
|---|---|---|---|---|---|

We see that even in this case the official inventory data provide much lower area fluxes for $CO_2$ and $CH_4$. The validity of our results can be confirmed if we consider the values of emission ratio (ER) which are widely used as a characteristic of the relative structure of emissions from a source. If we compare ERs estimated from our observational data (FTIR measurements during EMME campaign and in-situ routine observations of $CO_2$, CO and $CH_4$) and ERs derived from official inventory data, we can see that these values differ significantly from each other, see Table 2 (the former Table 5) in the paper. For example, the mean value of ER$_{CO/CO2}$ obtained from our observations varies from 5.9 to 6.2, at the same time the ER$_{CO/CO2}$ value estimated using official inventory data equals to 21. This difference in ER$_{CO/CO2}$ values obtained using "top-down" and "bottom-up" approaches could be explained by the underestimation of total $CO_2$ and $CH_4$ emission of St.Petersburg in the official inventory.

**6) Line 358: repetition of "April 25", please delete the second one.**

Repetition of "April 25" has been deleted.

**7) Equation 2: It is not clear what kind of wind speeds are taken for the consideration, please elaborate it.**

We added the following text:

> … where δV is the relative variation of the wind speed over a day estimated using HYSPLIT meteorological data,...

**8) Equation 2: you can determine the square root of the error terms instead of adding them**

The esteemed referee is perfectly right. The assumption of uncorrelated errors of input parameters should work well in our case. However, in order to be on the safe side we decided to present the estimation of the upper limit of the total error (completely correlated errors of wind and TC which are anticorrelated with the errors of L), therefore we added terms instead of using the square root of the sum of squared terms. In the original version of the manuscript we have already written: "The $\delta F$ values calculated in this way can be considered as an upper limit of the $F$ uncertainty."

**9) Figure 5: there is no unit for the color bar [0-25]. The river is drawn as blue, but it looks confusing because the blue color is also assigned to the color bar.**

In the revised version we changed the figure caption (Fig.3, former Fig.5):

> The HYSPLIT model output for each of the campaign days (10:00 UTC) used as the forecast of the megacity plume while planning the field campaign. The colour bar units for TC$_{NO2}$ are [0-25] $10^{15}$ cm$^{-2}$. The blue line in the southeast indicates the river Neva.

**10) Figure 7: you could show the scaled results instead. It will illustrate how the close the curves are to each other after the scaling process.**

Figure 7 (at present Fig. 5) in the original manuscript is showing the data after the scaling process. However, it was not indicated explicitly. In the revised version we give this information in the text of the article and in the figure caption:

The scaled results of the side-by-side measurements of XCO2, XCH4, and XCO by FTS#80 and FTS#84 on 12 April 2019 at the St. Petersburg observational site are presented in Fig. 5.

Figure 5: The scaled results of the side-by-side measurements of XCO2, XCH4, and XCO by FTS#80 and FTS#84 on 12 April 2019.

**11) Figure 8: It is not very clear from the description which paths you took for determining the effective path length, are these paths from different days? Please elaborate these further. Do you have only one effective path length for all the days for each meteorological data set (LOCAL, GDAS, and HYSPLIT)? If so, how the effective path lengths vary given by different meteorological data set?**

Figure 6 (former Fig.8) shows all the paths of our experiments, one path per day. They are all different, since the FTIR observation locations and the wind field change from day to day. In the original manuscript we announced in the figure caption that **for simplicity, the path lengths on the map are equal.** We agree that this phrase can be misleading. So, in the revised version the figure caption is changed:

"An example of linear backward paths (black straight lines, black dots show the downwind FTS locations) for the days of FTIR observations. The major land use classes are shown by different colours (blue for the water bodies, grey for the residential buildings/industrial areas, green for the parks and forests). The path lengths on the map are plotted equal only for illustrative purpose. In fact they are all different since the FTIR observation locations and the wind field change from day to day. Red line designates the official administrative boundary of the St. Petersburg agglomeration. Red "star" indicates the location of one of the major thermal power stations (TPS) located to the north of St. Petersburg. Map data © 2019 Yandex."

*Special notes:*

A number of typos have been found and corrected during the preparation of the revised version of the manuscript. All of them are not critical with respect to the results and conclusions.

We slightly rearranged the text by moving several small parts of the text to other places without any changes. The general structure of the article remained unchanged. This minor rearrangement was a result of revising the manuscript in accordance with the comments and suggestions of referees.

Maria Makarova
on behalf of all co-authors

---

## Author Comment (AC2) · 29 Sep 2020

Please see the supplement.

Please also note the supplement to this comment:
https://amt.copernicus.org/preprints/amt-2020-87/amt-2020-87-AC2-supplement.pdf

---

## Author Comment (AC3) · 29 Sep 2020

**The reply to the anonymous referee #2 (RC2)**

We are thankful to the referee for the very detailed analysis of our study. We agree with almost all comments and took them into account while preparing the revised version of the manuscript.

Below, the actual comments of the referee are given in **`bold courier font and blue colour`**.
The text added to the revised version of the manuscript is marked by red colour.

**`The paper is well written, with good language and nice, instructive graphs in most cases.`**

We are grateful to the referee for the positive assessment of our manuscript.

**`It is claimed that the objective of the paper is to provide emission numbers for Sankt Petersburg. However a significant, and in my mind, to big part of the paper describes the general methodology with complementary data. The abstract is rather long and detailed, and it should be made more concise with focus on the results. The main body is too detailed for a scientific paper: a) The Modis data is not relevant since it is not actively used, b) Remove nice photos of StPetersburg, c) In the introduction, there is a lot of explanation about different variants of obtaining windspeed and effective path, but this is not used in any significant extent in the results; this should be shortened.`**

We agree with the referee's statements. However, to our opinion, the details of the experiment can be helpful for better understanding and analysis of the obtained results. Therefore we decided not to remove the experiment details completely or to shrink the corresponding part of the manuscript, but to move these details to the Appendix. We made the following changes in the paper:
1) Figure 4 containing MODIS images has been moved to Appendix A;
2) Figure 3 has been removed from the revised version of the manuscript;
3) Part of the information on the EMME-2019 observation details (including Table 1), the overview of meteorological data for the days of the field campaign (including Table 2), and the analysis of wind speed and the wind direction for the days of the field campaign based on the different data sources (including Table 3) were also moved to Appendix A.

**`If I understand right, the methodology is the same as used in other campaigns (Berlin). In the introduction or elsewhere an overview about the other studies should be added with discussion on how comparable this study is to the other ones in terms of methodology and results . E.g. was effective path used by other studies.`**

Yes, the esteemed referee is right. In the introduction section of the original manuscript it was indicated: "The idea and the methodology of EMME experiment was based mainly on the studies by Hase et al. (2015), Ionov and Poberovskii (2015), Chen et al. (2016) and Viatte et al. (2017)". Following the advice of the referee we added the following text:

> … Chen et al. (2016) developed and used differential column methodology (downwind-minus-upwind column differences) for the evaluation of $CH_4$ emissions from dairy farms in the Chino area. Vogel et al. (2019) investigated the Paris megacity emissions of $CO_2$ by coupling the COCCON observations and atmospheric transport model framework (CHIMERE-CAMS) simulations.

… De Foy et al. (2007), Mellqvist et al. (2010), Johansson et al. (2014), and Kille et al. (2017) have applied mobile FTIR (Solar Occultation Flux technique) and mobile DOAS techniques to the large scale flux measurements.

**In Eq 1 you calculate the flux using total column (needed to get the right unit).**

We have made the necessary changes in section **4.2 Mass balance approach for area flux estimation**. The new version of this section which includes explicit indication of the units is given below:

The estimation of the area fluxes *F* was obtained on the basis of a mass balance approach implemented in the form of a one-box model. Box models are a widely used technique for the evaluation of urban and other emission fluxes (Hanna et al., 1982; Reid and Steyn, 1997; Arya, 1999; Zinchenko et al., 2002; Zimnoch et al., 2010; Strong et al., 2011; Hiller et al., 2014a; Chen et al., 2016; Makarova et al., 2018). In our case the following equation for the calculation of area flux was used:

$$F_j(t_k) = \frac{\Delta_{TC}(t_i) \cdot V_j(t_i)}{L_j(t_i)} \cdot k \,,$$

(2)

where *F* (unit: t km$^{-2}$ yr$^{-1}$) is the area flux, $t_i$ denotes the day of a single field experiment in the frame of the observational campaign. It should be emphasized that we used the steady-state approximation for all involved processes within the duration of a single field experiment, so $\Delta_{TC}$ (unit: molec. m$^{-2}$) is the mean TC difference between downwind (TC$_d$) and upwind ( TC$_u$) observations $\Delta_{TC}$=TC$_d$ - TC$_u$, *V* (unit: m sec$^{-1}$) is the mean wind speed, and *L* (unit: m) is the mean length of a path of an air parcel which goes through the urban territory of St. Petersburg agglomeration. The *k* coefficient converts the value of area flux from (unit: molec. m$^{-2}$ sec$^{-1}$ ) to (unit: t km$^{-2}$ yr$^{-1}$):

$$k = \frac{m_{gas} \cdot 31536 \cdot 10^6}{N_A} \,,$$

(3)

where $m_{gas}$ is the molecular mass of the target gas (unit: kg mol$^{-1}$), N$_A$ – Avogadro constant (unit: mol$^{-1}$), 31536·10$^6$ - the coefficient that converts the value of area flux from (unit: kg m$^{-2}$ sec$^{-1}$) to (unit: t km$^{-2}$ yr$^{-1}$). The data for the wind speed and the wind direction were taken from different sources of meteorological information (see section 4.3), and these sources are identified as *j* in Eq. 2. So, as a result, we obtained the set of values of *F*(*t*) for each of the meteorological data sources and for each day of field measurements. We note that below we will use the units t km$^{-2}$ yr$^{-1}$ for the values of *F*(*t*).

**You also introduce Xgas (I assume against total pressure). When do you use Xgas in the calculation? Is it only to show thing quantitatively? I assume in most cases te pressure is the same for up and downwind site ? Add in the text a definition of Xgas (not know for everyone) and describe what is your purpose here for showing it?**

Please, see the answer to this comment below (the answer to referee's comment to P8, row 128).

**For the wind used in the final results the authors rely on the Hysplit model, which in turn is based on a global model (NCEP) for the wind. The authors argue that the use of data from this model provides less variability in the final results. I argue that the wind variability is less**

**for the Hysplit data than for real measurements, since it is large domain model, and Hysplit will therefore artificially smooth the wind data. This should be beter discussed by the authors.**

We agree with the referee's statement that "**wind variability is less for the Hysplit data than for real measurements ... and Hysplit will therefore artificially smooth the wind data**". Nevertheless, to our opinion, HYSPLIT cannot be classified as a **"...large domain model...".** Following the advice of the referee, we presented our arguments in the extended discussion in the new version in Section 4.4:

> We selected HYSPLIT as one of the sources of the wind data since HYSPLIT is a widely used modelling system for the simulation of air parcel trajectories and the dispersion processes in the atmosphere which was tested in a lot of studies (HYSPLIT publications can be found using the following links: https://www.arl.noaa.gov/hysplit/hysplit-publications-meteorological-data-information/). Stein et al. (2007) noted that *Grid models are the best-suited tools to handle the regional features of these chemicals. However, these models are not designed to resolve pollutant concentrations on local scales. Moreover, for many species of interest, having reaction time scales that are longer than the travel time across an urban area, chemical reactions can be ignored in describing local dispersion from strong individual sources making Lagrangian and plume-dispersion models practical.* Stein et al. (2007) classify HYSPLIT as a local model which provides *the more spatially resolved concentrations due to local emission sources.* Therefore, for modelling of the evolution of the St.Petersburg plume we used the HYSPLIT model as a tool which perfectly fits the scale of considered atmospheric processes. This was also the reason for using HYSPLIT as the source of the wind data.

**The authors present their flux estimation based on modelled effective path. Such an excercise provides useful data but it is hard for the reader to understand how the data was produced and its errors, since the data represents a combination of measurements and model. I suggest presenting also the purely measured data based on a constant path. For the effective path the authors claim they made a land use analysis and they refer to a public web site but there little information given in the paper and it is hard for the reader to understand the assumptions made here. For instance, I am missing an explanation about what are the hypothesis about the detailed emission source categories and differentiation between species (CO2, CH4, NO2). The species above orginate from different emission source categories; e.g CH4 could partly come from the waterways (sewers and water canals) and pipelines rather than mobile and fixed combustion sources which are relevant for CO2 and NO2. This will make the effective path species dependent. The emissions from water ways could also be impacted by windspeed. I suggest adding a graph for the landuse model and include the model as complementary material for this paper.**

Addressing this issue, in the revised version of the paper we present the values of area flux calculated using constant path length and the description of the land use model. The results obtained with a constant path length are given in Table B1 (please see below) in the Appendix B.

**Table B1. Area fluxes for $CO_2$ (kt km$^{-2}$ yr$^{-1}$), $CH_4$ (t km$^{-2}$ yr$^{-1}$), CO (t km$^{-2}$ yr$^{-1}$) and $NO_x$ (t km$^{-2}$ yr$^{-1}$) obtained using constant path length approach.**

| Area flux | EMME | | In situ measurements |
|---|---|---|---|
| | (9 days) | (4 days) | |
| **1** | **2** | **3** | **4** |

| | | | |
|---|---|---|---|
| $CO_2$, kt km$^{-2}$ yr$^{-1}$ | 96 ± 25 | 99 ± 17 | 32 ± 27 |
| $CH_4$, t km$^{-2}$ yr$^{-1}$ | 151 ± 82 | 213 ± 57 | 95 ± 64 |
| CO, t km$^{-2}$ yr$^{-1}$ | 276 ± 117 | 385 ± 97 | 71 ± 40 |
| $NO_x$, t km$^{-2}$ yr$^{-1}$ | 74 ± 30 | - | - |

The land use model that was developed for the computation of the variable path length is presented in Fig.6 (former Fig.8):

In Fig. 6 these land use classes are shown in different colours: blue for the water bodies, grey for the residential buildings/industrial areas, green for the parks and forests. Effective path length is calculated as a sum of elementary paths through the urbanized grid pixels which contain residential buildings, industrial areas, and roads/highways. Pixels containing water bodies, swamps, and parks are excluded from the variable path calculations. Similar approach was implemented by Hase et al. (2015). The total urbanized area of the St.Petersburg agglomeration according to the developed land use classification occupies the area of 984 km$^2$ while the official area of the entire St.Petersburg is of 1439 km$^2$. The target gases can originate from different emission source categories, i.e. $CH_4$ could partly come from the waterways (sewers and water canals), wetlands and pipelines rather than mobile and point combustion sources which are relevant to CO, $CO_2$ and $NO_2$. The EMME-2019 was carried out during March-April when water bodies and earth surface were fully or partly covered by ice and snow (see Appendix A, Fig. A1), and soils were still frozen. Therefore we suggest that the $CH_4$ emission from the excluded pixels (water bodies, swamps, parks, and forests) was negligible in comparison to other anthropogenic sources (landfills, pipelines, etc.) which are distributed over the urbanized pixels.

We generally agree with the statement that "the emissions from water ways could also be impacted by windspeed" but this effect is not expected to be critical since water bodies were covered by ice and snow.

As it was mentioned above, for the revised version of the manuscript we computed the urbanized area of St.Petersburg agglomeration according to the land-use classification that was developed in order to estimate the effective path lengths. The total urbanized area of the agglomeration occupies 984 km$^2$ while the official area of the entire St.Petersburg is 1439 km$^2$. Therefore, the values of area fluxes for all gases ($CO_2$, $CH_4$, CO and $NO_2$) that were estimated using the official inventory data have been recalculated and, as a result became higher. Revised version of Table 1 (the former Table 4) is given below, corresponding changes are highlighted by yellow colour.

**Table 1. Area fluxes for $CO_2$ (kt km$^{-2}$ yr$^{-1}$), $CH_4$ (t km$^{-2}$ yr$^{-1}$), CO (t km$^{-2}$ yr$^{-1}$) and $NO_x$ (t km$^{-2}$ yr$^{-1}$) obtained during EMME-2019 and the flux estimates for St. Petersburg based on in situ measurements. The values previously reported in literature are also presented.**

| Area flux | EMME | | In situ measurements | Literature sources | |
|---|---|---|---|---|---|
| | (9 days) | (4 days) | | St. Petersburg | The world's cities |
| **1** | **2** | **3** | **4** | **5** | **6** |
| $CO_2$, kt km$^{-2}$ yr$^{-1}$ | 89 ± 28 | 85 ± 12 | 40 ± 30 | 31 (Serebritsky, 2018), 46 (EDGAR database, 2018) 6 (suburbs, Makarova, 2018) | 29 (London, O'Shea, 2014) 35.5 (London, Helfter, 2011) 12.8 (Mexico City, Velasco, 2005) 12.3 (Tokyo, Moriwaki and Kanda, 2004) 0.8 – 7.7 (Krakow, Zimnoch, 2010) 28.3 (Berlin, Hase, 2015) |
| $CH_4$, t km$^{-2}$ yr$^{-1}$ | 135 ± 68 | 178 ± 30 | 120 ± 80 | 25 (Serebritsky, 2018, 2019), 110 (Makarova, 2006), 44 (suburbs, Makarova, 2018) 32 (suburbs, Zinchenko, 2002) | 66 (London, O'Shea, 2014) 7 – 28 (Krakow, Zimnoch, 2010) |
| CO, t km$^{-2}$ yr$^{-1}$ | 251± 104 | 333 ± 103 | 90 ± 50 | 410 (Serebritsky, 2018, 2019), 390 (Makarova, 2011), 90 (suburbs, Makarova, 2018) | 106 (London, O'Shea, 2014) 1520 (Mexico City, Stremme, 2013) |
| $NO_x$, t km$^{-2}$ yr$^{-1}$ | 66 ± 28 | - | - | 69 (Serebritsky, 2018, 2019) | 63-252 (London, Lee, 2015) 13- 300 (Norfolk, Marr, 2013) |

**The NO2 DOAS data are explained very briefly wrt to methodology and results. Did you use the same methodology as for the other species, even though you measure in a full circle around town. I suggesting describing the methodology in a better way and results. Did you use the NO2 data to correct the FTIR measured data, if so clarify.**

A detailed description of our DOAS measurements can be found in the references provided in the manuscript (Ionov and Poberovskii 2012, Ionov and Poberovskii 2015, Ionov and Poberovskii 2017, Ionov and Poberovskii 2019). We would not like to increase the size of the manuscript by describing the methodology in every detail. However, as a response to the referee's comment, in the revised version we added the following text to Section 4:

Basically, the DOAS algorithm derives the $NO_2$ atmospheric content by fitting a reference $NO_2$ absorption cross-section to the measured zenith scattered radiance. The effective or slant column density (SCD) of $NO_2$ is retrieved in the 425-485 nm fitting window. SCD is converted then to vertical column density (VCD) by means of so-called air mass factor AMF (VCD=SCD/AMF), pre-calculated with a radiative transfer model (RTM). The spatiotemporal variations of stratospheric $NO_2$ are negligible compared to these in a polluted

troposphere. Consequently, the variations of $NO_2$ vertical column observed in the data of our mobile DOAS measurements are related to $NO_2$ pollution in the boundary layer (below ~1.5 km).

The primary purpose of mobile DOAS $NO_2$ measurements was a real-time verification of the pollution plume location with respect to the original HYSPLIT dispersion forecast. By means of this approach, the actual evolution of plume was monitored to adjust the FTIR field measurement positions, if necessary. We do mention this in the manuscript: "The real-time corrections of the FTIR operation sites were performed depending on the *actual evolution of the megacity $NO_x$ plume as detected by the mobile DOAS observations*" (lines 35-36 of the Abstract, orig. version), and "The concept of EMME is based on remote measurements of the total column amount of $CO_2$, $CH_4$ and CO from two mobile platforms located inside and outside the city plume (usually at upwind and downwind locations on the opposite sides of the city of St. Petersburg) combined with the *mobile circular measurements of tropospheric column amount of $NO_2$ from the third mobile platform moving in a non-stop mode, the latter measurements are used for the real-time control of the megacity plume evolution*" (beginning of Section 2, orig. version). Generally, the DOAS measurements confirmed the HYSPLIT forecast. However, on one day of experiment this was not the case, and the FTIR measurements location was timely corrected according to the data of DOAS observations. This is mentioned at the end of Section 3.1, lines 217-221, orig. version.

The referee is right, the methodology of mass balance approach was applied to estimate $NO_x$ flux in exactly the same way as it was done for all other species ($CO_2$, $CH_4$ and CO). We do mention this in the manuscript: "The summary of the EMME-2019 results and the comparison with the flux estimates for St. Petersburg based on in situ measurements, as well as independent literature data, are presented in Table 4 (orig. version) for $CO_2$, $CH_4$, CO *and $NO_x$ (the latter were derived from mobile DOAS measurements of tropospheric $NO_2$ in the vicinity of upwind and downwind FTIR observations)*" (line 401-404 of Section 5.1, orig. version). Indeed, much more data of $NO_2$ measurements is available from our circular DOAS observations, but its interpretation is a subject of separate study and is beyond the scope of the manuscript under review. Finally, an answer to another referee's question here: no, we did not use the $NO_2$ data to correct the FTIR measured data.

**The treatment of uncertainties is all based on the obtained/measured variability of the parameters used to calculate the flux (total column, effective path and wind).**
**In my mind this is an assessment of the random uncertainty. However there is no mentioning of systematic errors of any of these parameters. Please add a discussion about this and change absolute uncertainties to random uncertainty.**

The following discussion was added in the paper:

To evaluate systematic error of the area flux ($\delta F_{sys}$) we should first estimate the systematic errors $\delta L_{sys}$, $\delta V_{sys}$ and $\delta \Delta TC_{sys}$ of corresponding parameters $L$, $V$ and $\Delta TC$ in Eq.2. In contrast to $\delta L_{sys}$ and $\delta V_{sys}$, the contribution of systematic component of $\delta \Delta TC_{sys}$ into $\delta F_{sys}$ is negligible. This is due to the high accuracy of the COCCON observations of gas columns which are calibrated against WMO scale. In Eq. 2 we use an assumption that an air parcel moves along a straight line but obviously this is not true. For the whole ensemble of HYSPLIT trajectories simulated for all days of the city campaign we calculated the maximum relative difference between the true lengths of HYSPLIT trajectories and our straight line approximations of $L$. This value equals to ~4% which is considered as an estimation of the relative systematic error $\delta L_{sys}$. According to the information on wind speed observed during the field campaign (see Appendix A, Table A3), the mean relative difference between HYSPLIT and GDAS data on wind speed is of 14±22%. Hence, the estimation of the systematic error of area flux $\delta F_{sys}$ due to the systematic errors of all parameters in Eq.2 gives the value 18%.

**In the CO2 and CO data there is a factor of two difference between the column measured data and the one measured by in situ data. This is explained by the fact that the CO2 and CO emissions are released from high chimneys (200m). However the mixing layer should be several hundred meters (at minimum) at solar conditions and the pollutants should therefore well mixed at some distances from the chimney (>1 km). This was also supported by kite measurements. In addition a considerable portion of the CO2 should come from transport sector. The discussion should be improved on this topic.**

We agree with the referee that this issue requires some more discussion. Taking into account that this topic is specific, we put the extended discussion in Appendix C:

**Appendix C: Comments on transport of the pollutants from elevated sources**

We illustrate transport of the pollutants from elevated sources with a HYSPLIT simulation (see Fig. C1). We selected one of the days of EMME (April 16, 2019) and simulated the $CO_2$ emission from a 180-meter chimney of the thermal power station mentioned above in the main text of the article. The plot presents a 34-hour trajectory of the mass-weighted $CO_2$ plume position (the centroid of the plume) on the geographical map (top panel) and using the altitude scale (bottom panel). One can see that the plume centroid starts its movement from the chimney location at ~180 m altitude (12:00 of April 15) and raises up to ~500 m in one hour; then it does not fall below the level of ~350 m during its "flight" length of more than 300 km. The detailed analysis of respective vertical profiles of $CO_2$ concentration shows its maximum at ~500 m, being 1.2 times higher than that on the surface at start and 3.6 times higher than that on the surface at the end of the plume trajectory. Thus, the probability to register high concentrations corresponding to the centroid of the plume by surface-based observations can be estimated as very low. Moreover, polluted air mass from a chimney is more likely to rise up, rather than descend to the ground due to two reasons: (1) the vertical velocity of the air pollution jet emitted from a chimney can be rather high; (2) the temperature of a plume released from the chimney is usually significantly higher than the temperature of the ambient air causing the buoyancy effect.

Elevated air sampling using kite launches was performed only twice during the EMME campaign, therefore the results of these kind of measurements could not be considered as a reliable confirmation of the absence of elevated plumes. The presence of the elevated plumes of CO and $CO_2$ could be also confirmed by the following evidence. The comparison of the values of area fluxes ($F$, see Table 1) estimated using in-situ measurements (column #4) and FTIR observations (column #2 and #3) shows that for $CH_4$ which sources are mainly located on the ground surface we obtain significantly lower difference in corresponding $F$ values than for CO an $CO_2$.

[Figure]

NOAA HYSPLIT MODEL (mass-weighted centroid position of CO2 plume)
Forward trajectory starting at 1200 UTC 15 Apr 19
GFSG Meteorological Data

**Figure C1: Evolution of the mass-weighted centroid position of the CO2 plume taken as an example (see text).**

**Specific comments**

**P3: Row 83: When making refence to other studies it would be relevant to add similar large scale measurements by mobile FTIR (Solar Occultation Flux technique) and mobile DOAS which has been applied fo large scale flux measurements for at least decade by now : e.g. 1. de Foy, et al., (2007) Modelling constraints on the emission inventory and on vertical dispersion for CO and SO2 in the Mexico City Metropolitan Area using Solar FTIR and zenith sky UV spectroscopy. Atmospheric Chemistry And Physics 7, pp. 781-801. DOI: 10.5194/acp-7-781-2007. 2. Mellqvist, et al., (2010) Measurements of industrial emissions of alkenes in Texas using the solar occultation flux method. Journal of Geophysical Research - Atmospheres 115. DOI: 10.1029/2008JD011682. 3. Johansson, J., et al. (2014) Emission measurements of alkenes, alkanes, SO2, and NO2 from stationary sources in Southeast Texas over a 5 year period using SOF and mobile DOAS. Journal of Geophysical Research-Atmospheres 119, no. 4, pp. 1973-1991. DOI: 10.1002/2013jd020485. 4. Johansson, et al. (2014) Quantitative measurements and modeling of industrial formaldehyde emissions in the Greater Houston area during campaigns in 2009 and 2011. Journal of Geophysical Research-Atmospheres 119, no. 7, pp. 4303-4322. DOI:10.1002/2013JD020159. 5. Kille N, et al, The CU Mobile Solar Occultation Fluxinstrument, AMT, 10, 373-392, 2017**

The following text has been added in the introduction section:

… Chen et al. (2016) developed and used differential column methodology (downwind-minus-upwind column differences) for the evaluation of $CH_4$ emissions from dairy farms in the Chino area. Vogel et al. (2019) investigated the Paris megacity emissions of $CO_2$ by

coupling the COCCON observations and atmospheric transport model framework (CHIMERE-CAMS) simulations."

……………

"… De Foy et al. (2007), Mellqvist et al. (2010), Johansson et al. (2014), and Kille et al. (2017) have applied mobile FTIR (Solar Occultation Flux technique) and mobile DOAS techniques to the large scale flux measurements.

**P 5, row 121: You claim that the DOAS measures tropospheric columns. Please elaborate in a few sentences what is actually measured, even though you refer to previous studies. Are you using multiaxis measurements to derive absolute columns or is it differential columns assuming that the upwind measurements is free from troposheric NO2, and hence that the differential measurements corresponds to the tropospheric absolute column.**

In the revised version of our manuscript we added a text with some more details of our DOAS measurements (see above). We are not using multiaxis (or MAX-DOAS) observations. Our DOAS measurements are just zenith-sky, and we specify that in the manuscript.

**P5, row 132. Add references from other places on mobile DOAS, e.g. Johansson, M et al., Mobile mini-DOAS measurement of the outflow of NO2 and HCHO from Mexico city, ACP, 9(15):5647-5653, 2009. Rivera, C. et al., (2010) Quantification of NO2 and SO2 emissions from the Houston Ship Channel and Texas City industrial areas during the 2006 Texas Air Quality Study. Journal of Geophysical Research - Atmospheres 115. DOI: 10.1029/2009JD012675.**

In the revised version we added the following sentence and significantly expanded the list of relevant references:

In general, such observations have been proved to be an efficient technique to derive the anthropogeinc $NO_x$ flux in many studies worldwide (see e.g., Johansson et al., 2008, Rivera et al., 2009, Johansson et al., 2009, Rivera et al., 2010, Ibrahim et al., 2010, Shaiganfar et al., 2011, Wang et al., 2012, Shaiganfar et al., 2015, Wu et al., 2017, Shaiganfar et al., 2017).

**P6, row 171: This sentence is unclear rewrite it. For instance Table 1 presents daily information …**

In the revised version we added the following text:

Table A1 (see Appendix A) presents daily information on the location of FTIR spectrometers during the campaign, FTIR spectrometer identifier, number of bags of air samples, flight of a kite and air sampling altitude.

**P8, row 128: Define Xgas (is it against pressure?) and motivate why you introduce this. Would it not be more appropiate to compare total columns instead of Xgas since TC is the ones used for the flux.**

For the cross-calibration of the EM27/SUN spectrometers we used $XCO_2$, $XCH_4$, and XCO values as strongly recommended in the special study by Frey et al. (2015). To define Xgas, we added the following text:

The ratio of the target gas TC to the retrieved $O_2$ TC which is suggested to be known and constant, gives us the column-averaged dry-air mole fraction ($X_{gas}$) of the target gas (Wunch et al., 2011; Frey et al., 2015):

$$Xgas = 0.2095 \frac{TCgas}{TC_{O2}} = \frac{TCgas}{TCdry\ air}, \qquad (1)$$

where $X_{gas}$ - column-averaged dry-air mole fraction of the target gas (unit: dimensionless quantity), $TC_{gas}$ – total column of the target gas (unit: molec. m$^{-2}$), $TC_{O2}$ - total column of $O_2$ (unit: molec. m$^{-2}$), $TC_{dry\ air}$ – dry air total column (unit: molec. m$^{-2}$). Using Xgas helps to reduce the effect of various possible systematic errors (Wunch et al., 2011). To provide the compatibility of EM27/SUN measurements to WMO scale and for consistency reasons, the retrieval software used for processing the EM27/SUN spectra also performs a post-processing (Frey et al., 2015). Finally, we had at our disposal both the TCgas and Xgas for each day of measurements at each observational location.

**P8, row 232: The comparions between the two spectrometers is very convincing. Nevertheless, it only shows how the spectral properties of two spectrometers influences the statistical error of the measurements. Please comment how this information was used.**

After cross-comparison procedure we used obtained regression parameters to scale the data. The result after the scaling process is shown in Figure 5. We explain it in the revised version:

The calibration factors obtained as a result of side-by-side comparison were used to convert $XCO_2$, $XCH_4$, and XCO measured by spectrometer #80 to the scale of spectrometer #84. The results of cross-calibration help to avoid an additional source of systematic error in the estimation of area fluxes.

**P 9, 244: I think this section should be more detailed wrt the spectroscopy. At least a couple of general sentences for how te retrieval is done and if there are interfering species etc could be helpful,**

In the revised version we added the following text in section **4.1 FTIR and DOAS data processing**:

...For the retrievals of the total columns of $O_2$, $CO_2$, CO, $H_2O$, and $CH_4$, the spectral regions recommended by Frey et al. (2019) and Hase et al. (2016) were taken. We present these intervals in the respective order: 7765 – 8005 cm$^{-1}$ (the main interfering gases are $H_2O$, HF, $CO_2$), 6173 – 6390 cm$^{-1}$ (the main interfering gases are $H_2O$, HDO, $CH_4$), 4210 – 4320 cm$^{-1}$ (the main interfering gases are $H_2O$, HDO, $CH_4$), 8353 – 8463 cm$^{-1}$, and 5897 – 6145 cm$^{-1}$ (the main interfering gases are $H_2O$, HDO, $CO_2$). The EM27/SUN spectrometer has low spectral resolution of 0.5 cm$^{-1}$. Therefore the TCs are derived from the FTIR spectra by scaling of a priori profiles of target gases (Frey et al., 2019).

*Special note:*

A number of typos have been found and corrected during the preparation of the revised version of the manuscript. All of them are not critical with respect to the results and conclusions.

We slightly rearranged the text by moving several small parts of the text to other places without any changes. The general structure of the article remained unchanged. This minor rearrangement was a result of revising the manuscript in accordance with the comments and suggestions of referees.

Maria Makarova
on behalf of all co-authors